# Slight compositional variation-induced structural disorder-to-order transition enables fast Na$^+$ storage in layered transition metal oxides

Yuansheng Shi [1], Pengfeng Jiang[1], Shicheng Wang[1], Weixin Chen[1], Bin Wei [1], Xueyi Lu[1], Guoyu Qian [1], Wang Hay Kan[2,3], Huaican Chen[2,3], Wen Yin[2,3], Yang Sun[1] & Xia Lu [1] ✉

The omnipresent Na$^+$/vacancy orderings change substantially with the composition that inevitably actuate the ionic diffusion in rechargeable batteries. Therefore, it may hold the key to the electrode design with high rate capability. Herein, the influence of Na$^+$/vacancy ordering on Na$^+$ mobility is demonstrated firstly through a comparative investigation in P2-Na$_{2/3}$Ni$_{1/3}$Mn$_{2/3}$O$_2$ and P2-Na$_{2/3}$Ni$_{0.3}$Mn$_{0.7}$O$_2$. The large zigzag Na$^+$/vacancy intralayer ordering is found to accelerate Na$^+$ migration in P2-type Na$_{2/3}$Ni$_{1/3}$Mn$_{2/3}$O$_2$. By theoretical simulations, it is revealed that the Na$^+$ ordering enables the P2-type Na$_{2/3}$Ni$_{1/3}$Mn$_{2/3}$O$_2$ with higher diffusivities and lower activation energies of 200 meV with respect to the P3 one. The quantifying diffusional analysis further prove that the higher probability of the concerted Na$^+$ ionic diffusion occurs in P2-type Na$_{2/3}$Ni$_{1/3}$Mn$_{2/3}$O$_2$ due to the appropriate ratio of high energy ordered Na ions (Na$_f$) occupation. As a result, the interplay between the Na$^+$/vacancy ordering and Na$^+$ kinetic is well understood in P2-type layered cathodes.

For the "carbon peaking and carbon neutrality" goals, it is highly expected to increase the ratio of renewable energy in our modern industrial society[1]. However, the intermittent renewable energy currently causes discontinuous inputs to the State Grid. Rechargeable batteries can be therefore supplemented as a reliable power station stably connecting the clean energies with the consumers to improve the State Grid resilience[2]. Limited by Li resources and costs, in recent years, the "beyond Li-ion battery (LIB)" technologies, such as the sodium-ion battery (SIB) have developed rapidly because of their economic advantage and good electrochemical performance. However, the present SIBs cannot meet the alternative high-rate cycling stationary power in rapid succession. From a material's perspective of

view, the problem lies mainly in electrode materials, including the lack of alternative candidates and the ambiguous mechanism between Na diffusion kinetics and atomic structure upon battery cycling[3]. Hence, material design and exploration becomes the dominant work in building the better practical SIBs for cleaning energy, life and the future.

In the available SIB cathodes, the layered oxides as Na$_x$TMO$_2$ (TM = transition metals) with the typical 2D Na$^+$ diffusion kinetics are the up-and-coming candidates due to their excellent electrochemical activity, high energy density and low costs with respect to the polyanionic materials, the Prussian blue analogs and the organics[4,5]. According to the coordination of Na$^+$ and the spatial arrangements of

[1]School of Materials, Sun Yat-sen University, Shenzhen 518107, People's Republic of China. [2]Spallation Neutron Source Science Center, Dongguan 523803, People's Republic of China. [3]Institute of High Energy Physics, Chinese Academy of Sciences, Beijing 100049, People's Republic of China. ✉e-mail: luxia3@mail.sysu.edu.cn

$MO_6$ octahedra, the sodium-based layered oxides are classified into many types, such as the popular P2, P3, O2 and O3 structures, which are previously established by Delmas et al. in the early 1980s[6]. The symbols of "P" or "O" refer to the coordination environments of interstitial cations, e.g., $Na^+$ or $Li^+$, where they are prismatically or octahedrally coordinated to the neighboring oxygen ions. The numerical 2 or 3 suggests the repeated $MO_6$ slabs in one cell unit[6,7]. At present, the prismatic sublattice arrangements in those "P" phases make them more accessible for faster $Na^+$ migration in general with respect to that in "O" phases[8]. However, the Na diffusion kinetics are complex and probably the case-by-case issues in different materials. The observed Na diffusion (rate capability) is associated with many factors, such as the Na slab ordering/disordering[9], charge ordering[10], phase transition[11], electron transport[12], and the correlated ionic motions[13], which are entangled with each other. Among these essential roles, the influence of the $Na^+$/vacancy disorder-to-order transition on the Na kinetics seems to be an important content because of the significant challenges of finding a suitable material system to figure its contribution out in P2-type layered oxides[10]. The reason is that many previous works just emphasized on decoding the contributions from transition metals within TM layer by introducing heteroatoms[14,15].

Hence, two layered sodium oxides are selected to check this $Na^+$ diffusion kinetics in P type layered cathodes. The first one is the P2-$Na_{2/3}Ni_{1/3}Mn_{2/3}O_2$, which has been previously reported to be in-plane $Na^+$/vacancy ordering with the intralayer TM honeycomb ordering. The other one is the P2-$Na_{2/3}Ni_{0.3}Mn_{0.7}O_2$, which undergoes disrupted in-plane $Na^+$/vacancy ordering without changing the TM honeycomb ordering after structural perturbation. It is indicated that substantially different $Na^+$ mobility results from the $Na^+$/vacancy disorder-to-order transition in these two samples, since the challenges even pose at the atomic level to simulate the in-plane $Na^+$/vacancy disordering in P2-$Na_{2/3}Ni_{0.3}Mn_{0.7}O_2$. Therefore, the precise control in the synthesis of $Na_{2/3}Ni_{1/3}Mn_{2/3}O_2$ polymorphs opens the possibility of directly comparing the $Na^+$ diffusion kinetics in P2/P3 phases with the exact stoichiometric ratios. It is reported that there are the Ni/Mn superlattice orderings in TM layers within both P2- and P3-$Na_{2/3}Ni_{1/3}Mn_{2/3}O_2$ phases, and it simultaneously accompanies with the in-plane large zigzag (LZZ) $Na^+$/vacancy ordering inside the P2-$Na_{2/3}Ni_{1/3}Mn_{2/3}O_2$. In contrast, no preferential in-plane Na ordering is reported in P3-$Na_{2/3}Ni_{1/3}Mn_{2/3}O_2$[16,17]. Hence, the chances emerge to elucidate the influence of in-plane $Na^+$/vacancy ordering on its diffusion kinetics in P2-type layered oxides. Consequently, in the conjugation of electrochemistry with density functional theory (DFT) simulations, the concerted $Na^+$ diffusion kinetics are disclosed in the LZZ ordering in P2-type layered oxides as a new guidance in materials' design of high-rate SIBs.

## Results

### Disrupting the in-plane $Na^+$/vacancy ordering

In the available P2 layered oxides, previous works have already indicated preliminarily the $Na^+$ LZZ ordering and the TM honeycomb ordering, e.g., in P2-$Na_{2/3}Ni_{1/3}Mn_{2/3}O_2$ cathode using X-ray/neutron powder diffraction (NPD)[9,17]. While, the in-plane $Na^+$/vacancy ordering is disrupted as shown in Fig. 1a by varying the Ni/Mn ratios from 1/2 to 3/7, where the superstructure peaks between 25 and 30° correspond to the average distance of the adjacent sodium ions with the LZZ ordering in P2-$Na_{2/3}Ni_{1/3}Mn_{2/3}O_2$ (details see Figs. S1, S2 and Tables S1, S3). Moreover, in contrast to the similar X-ray scattering factors of Ni and Mn ions, the strong neutron scattering length contrast of Ni and Mn enables the NPD to be an effective measurement to detect the Ni−Mn intralayer ordering as shown in Fig. 1b. The NPD patterns (batch of Bank 4) present that the diffraction peaks in red shadow result from the Na LZZ ordering within the Na slabs, while the peaks in yellow shadow are assigned to the Ni−Mn honeycomb ordering within the TM layers. The good agreement between the experimental NPD data and

the proposed ordering structures is demonstrated in Fig. S3 (refer to Supplementary Notes 1, 2). Then, the refined experimental NPD pattern (batch of Bank 6) indicated that the as-prepared phase pure P2 samples present the space group of $P6_322$ with a clearly TM honeycomb ordering as shown in Fig. 1c, d (crystallographic parameters see Tables S2, S4).

The TEM images of the as-prepared P2-$Na_{2/3}Ni_{1/3}Mn_{2/3}O_2$ and P2-$Na_{2/3}Ni_{0.3}Mn_{0.7}O_2$ cathodes further demonstrate the aforementioned structural differences. These two samples both display the schistose particles in 1–5 μm as shown in Fig. S4. The homogeneous distribution of Na, Ni, and Mn elements is observed in the both of two samples as shown in Fig. S5. From the selected area electron diffraction (SAED) patterns, the $Na^+$ LZZ ordering in P2-$Na_{2/3}Ni_{1/3}Mn_{2/3}O_2$ is captured at the atomic level as shown in Fig. 2a, in consistence with the XRD and NPD results. Moreover, the Ni-Mn ordering with only intralayer ($\sqrt{3} \times \sqrt{3}$)-$R30°$ superstructure is disclosed in P2-$Na_{2/3}Ni_{0.3}Mn_{0.7}O_2$, which is denoted by the weak SAED spots in white circles as shown in Fig. 2b (details see Fig. S6). As a matter of fact, the SAED results indicate that the TM ordering phenomenon can maintain in a range of Ni/Mn ratios. The inset STEM HAADF image clearly shows the atomic arrangement of Ni and Mn ions with honeycomb ordering along the [001] direction (see Fig. S7). Raman spectra are also conducted to investigate the spatial distribution of $Na^+$ in P2-$Na_{2/3}Ni_{1/3}Mn_{2/3}O_2$ and P2-$Na_{2/3}Ni_{0.3}Mn_{0.7}O_2$ as shown in Fig. 2c. In terms of the $P6_3/mmc$ space group (typically for P2 type layered oxides), the Raman active modes are the $\Gamma$(Raman, optic) = $A_{1g} + 3E_{2g} + E_{1g}$ bands, where the $A_{1g}$ and $E_{1g}$ modes are only correlated with the oxygen vibrations (O site: 4f). The $E_{2g}$ modes are associated with the vibrations of $Na^+$ (Na sites: 2b, 2d). Referring to the polarized Raman results by Qu et al.[18], the Raman peak at 460 cm$^{-1}$ is identified as the $E_{2g}$ mode, and another one at 576 cm$^{-1}$ is the $A_{1g}$ mode of $Na_xCoO_2$, which corresponds to the Raman peaks at 478 cm$^{-1}$($E_{2g}$) and 588 cm$^{-1}$($A_{1g}$) with the apparent shifts in the as-prepared P2-$Na_{2/3}Ni_{1/3}Mn_{2/3}O_2$ and P2-$Na_{2/3}Ni_{0.3}Mn_{0.7}O_2$ samples here. Of particular importance is that the broadened shoulder peak at ~640 cm$^{-1}$ in P2-$Na_{2/3}Ni_{0.3}Mn_{0.7}O_2$ samples is also detected in P2-$Na_{2/3}Al_{1/24}Ni_{7/24}Mn_{2/3}O_2$ and P2-$Na_{2/3}Co_{1/6}Ni_{1/6}Mn_{2/3}O_2$ (Figs. S12, S13), which might connect with the Na ions disordered distribution or the possible sublattice formation[19,20]. This broadened Raman spectra is an indirect response to the structural disordering/ordering formations from the refined XRD/NPD results. After all, the Na slab ordering is disrupted, while the TM honeycomb ordering is kept in $Na_{2/3}Ni_{0.3}Mn_{0.7}O_2$ material. Consequently, there is an ideal controllable experiment to individually identify the influence of $Na^+$/vacancy ordering on the $Na^+$ migration, structural stability, and electrochemical performance through the comparative study of the as-prepared P2-$Na_{2/3}Ni_{1/3}Mn_{2/3}O_2$ and P2-$Na_{2/3}Ni_{0.3}Mn_{0.7}O_2$.

### The $Na^+$ diffusion kinetics

Intriguingly, the μm-sized P2-$Na_{2/3}Ni_{1/3}Mn_{2/3}O_2$ single crystal demonsrates substaintily different electrochemical kinetics with respect to that of P2-$Na_{2/3}Ni_{0.3}Mn_{0.7}O_2$. Noting that voltage range of 2.0–4.0 V was selected in this case to exclude out the high voltage phase transition process (P2-O2)[21,22] plus the oxygen activities[23], as well as the low voltage of Mn redox[24]. The P2-$Na_{2/3}Ni_{0.3}Mn_{0.7}O_2$ cathode with $Na^+$/vacancy disordering delivers the rate capability of 89.4 mA h g$^{-1}$ at 0.2 C (30 mA g$^{-1}$), 78 mA h g$^{-1}$ at 1.0 C and 35.4 mA h g$^{-1}$ at 5.0 C, which are overwhelmingly inferior to that of P2-$Na_{2/3}Ni_{1/3}Mn_{2/3}O_2$ with $Na^+$/vacancy ordering as shown in Fig. 2d (details see Fig. S8). As shown in Fig. 2e, f, the CV tests are carried out at various sweeping rates to obtain the apparent chemical diffusion coefficients of $Na^+$ under the same conditions. For a homogeneous system, according to the Randles−Sevcik equation[25], $i_p = 2.69 \times 10^5\ n^{3/2}C_0AD^{1/2}v^{1/2}$, where the $i_p$ represents the peak currents, $n$ is the number of electrons per reaction species, the $C_0$ is the concentration of $Na^+$ in the lattice, $A$ is the area of electrode, $D$ is apparent $Na^+$ diffusion coefficient, $v$ stands for the scan

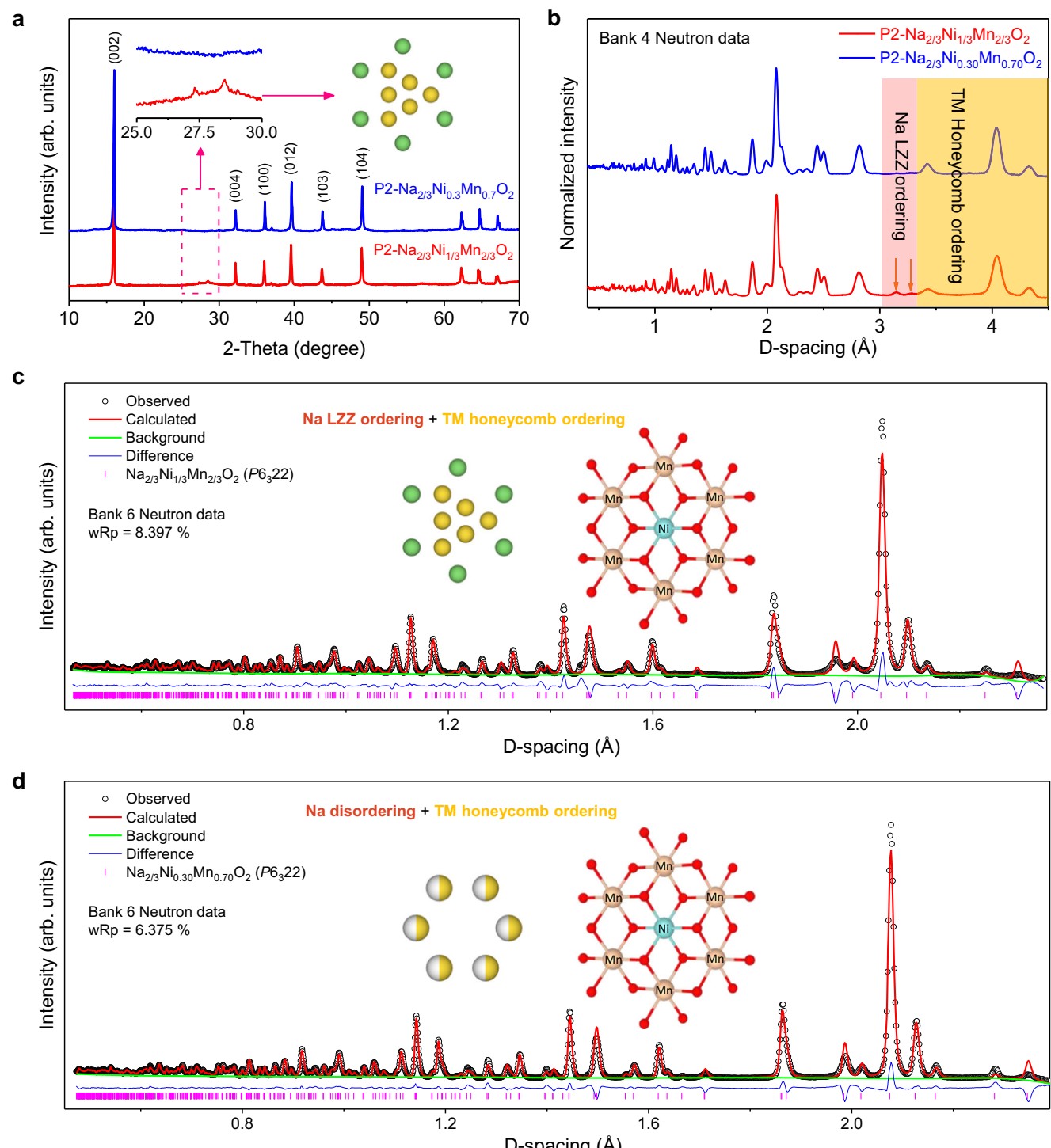

**Fig. 1 | Disrupting the in-plane Na⁺/vacancy ordering in layered P2 phase under perturbation.** **a** XRD patterns of the as-prepared P2-Na$_{2/3}$Ni$_{1/3}$Mn$_{2/3}$O$_2$ and P2-Na$_{2/3}$Ni$_{0.3}$Mn$_{0.7}$O$_2$ sintered at 950 °C for 15 h, including the loss of in-plane Na⁺/vacancy ordering in P2-Na$_{2/3}$Ni$_{0.3}$Mn$_{0.7}$O$_2$. **b** NPD patterns of the as-prepared P2-Na$_{2/3}$Ni$_{1/3}$Mn$_{2/3}$O$_2$ and P2-Na$_{2/3}$Ni$_{0.3}$Mn$_{0.7}$O$_2$ samples. Refined NPD patterns for **c** P2-Na$_{2/3}$Ni$_{1/3}$Mn$_{2/3}$O$_2$ and **d** P2-Na$_{2/3}$Ni$_{0.3}$Mn$_{0.7}$O$_2$ materials, wherein the experimental (black circles) and calculated (solid red line), the Bragg reflection peaks (magenta and solid purple ticks) and the difference curve (blue line) are shown, respectively.

rate. The parameters $n$, $C_0$, and $A$ are roughly equivalent in these two P2 structures. It can be seen that from Figs. S8, S9, the peak currents ($i_p$) and the square root of the scan rate ($v^{1/2}$) are positively correlated upon sodiation and desodiation. By fitting to the Randles−Sevcik equation, the average slope of P2-Na$_{2/3}$Ni$_{1/3}$Mn$_{2/3}$O$_2$ is ~2 times larger than that of P2-Na$_{2/3}$Ni$_{0.3}$Mn$_{0.7}$O$_2$, which means that the Na⁺ diffusion coefficient is significantly reduced as the Na LZZ ordering disappears

(fitting results refer to Table S5). This is consistent with the rate performance results as shown in Fig. 2d. Moreover, the optic bandgaps of P2-Na$_{2/3}$Ni$_{1/3}$Mn$_{2/3}$O$_2$ and P2-Na$_{2/3}$Ni$_{0.30}$Mn$_{0.70}$O$_2$ are equivalent roughly as shown in Fig. S10, which probably means the exclusion of electronic transport influence on the rate capability.

In addition, one more deviation of the P2-Na$_{2/3}$Ni$_{1/3}$Mn$_{2/3}$O$_2$ phase is the minor Al doped P2-Na$_{2/3}$Al$_{1/24}$Ni$_{7/24}$Mn$_{2/3}$O$_2$ (structural

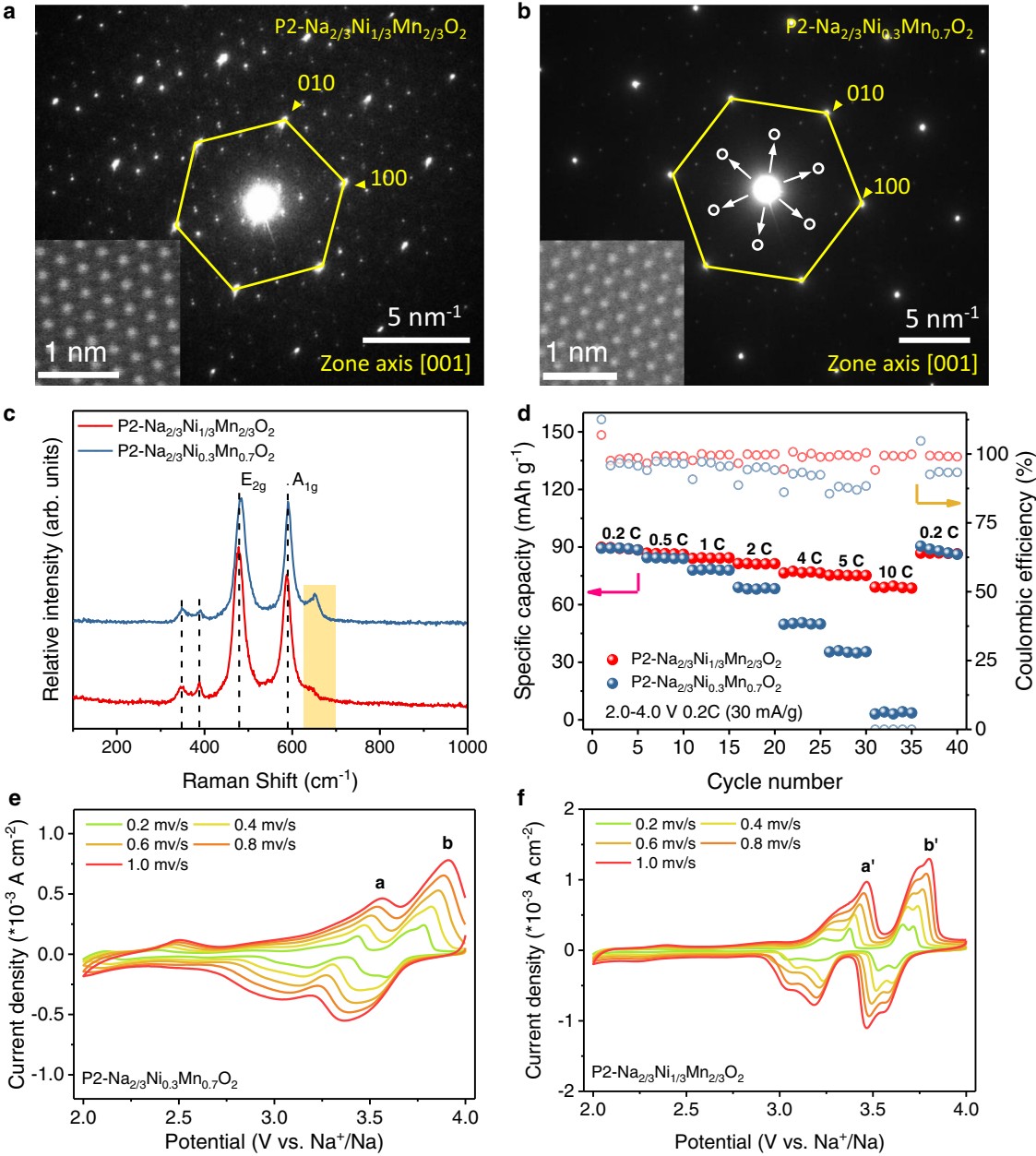

**Fig. 2 | TEM and electrochemistry of the layered P2 phases.** SAED patterns of **a** P2-Na$_{2/3}$Ni$_{1/3}$Mn$_{2/3}$O$_2$ and **b** P2-Na$_{2/3}$Ni$_{0.3}$Mn$_{0.7}$O$_2$ from [001] zone axis, and the corresponding STEM-HAADF images along the [001] direction are displayed at the insets. **c** Raman spectra, **d** rate performance and **e, f** CV curves of P2-Na$_{2/3}$Ni$_{1/3}$Mn$_{2/3}$O$_2$ and P2-Na$_{2/3}$Ni$_{0.3}$Mn$_{0.7}$O$_2$ samples at different scanning rates between 2.0 and 4.0 V.

perturbation by Al doping) is prepared to recheck the electrochemical performance difference that relates with the Na ion occupations. This tiny composition modulation does not cause significant changes in the shape of the electrochemical curve, but disrupts the Na slab LZZ ordering as shown in Fig. S11, S12. The subsequent rate capability test shows similar results as disclosed in the P2-Na$_{2/3}$Ni$_{0.3}$Mn$_{0.7}$O$_2$ (Figs. S8, S9, S12 and Table S5). Furthermore, the absence of superlattice peaks in another composition as the P2-Na$_{2/3}$[Co$_x$Ni$_{1/3-x}$Mn$_{2/3}$]O$_2$ by introducing 1/6 Co into the transition layers also means the disappearance of the ordered Na ion and transition metal cation arrangements as reported[16] (see Fig. S13 and Supplementary Note 3). The electrochemical performance of P2-Na$_{2/3}$Co$_{1/6}$Ni$_{1/6}$Mn$_{2/3}$O$_2$ is also inferior to that of P2-Na$_{2/3}$Ni$_{1/3}$Mn$_{2/3}$O$_2$ at high current density. Hence, the fast Na$^+$ diffusion is strongly correlated with the Na$^+$/vacancy ordering, and probably is accelerated by it. As a matter of fact, comprehensive details can be hardly achieved either from experimental characterizations or

theoretical calculations upon the facilitation of Na LZZ ordering on its diffusion kinetics due to the partial Wyckoff occupation of the face shared (Na$_f$, 2b) and the edge shared (Na$_e$, 2d) Na ions locating between the TMO$_6$ slabs, which is, in turn, a perquisition for forming the Na$^+$ LZZ ordering in P2-type oxides[9]. Note that in contrast to the P2-Na$_{2/3}$Ni$_{0.3}$Mn$_{0.7}$O$_2$, the anodic/cathodic peak splits are observed between 3 and 3.8 V in P2-Na$_{2/3}$Ni$_{1/3}$Mn$_{2/3}$O$_2$ electrode as shown in Fig. 2f, which might correspond to an order-disorder phase transition process[9].

To minimize the influence of Na composition variations on its spatial ordering, the same stoichiometric P3-Na$_{2/3}$Ni$_{1/3}$Mn$_{2/3}$O$_2$ layered cathode is further employed to examine the Na$^+$ kinetics. In P3 phase, the Na$^+$ locate in the same prismatic environment and two adjacent intralayer Na prismatic sites are energetically equivalent in the (*ab*) plane, with the only different coordination that is resulted from the different stacking sequence of MO$_6$ slabs[26,27]. Therefore, the

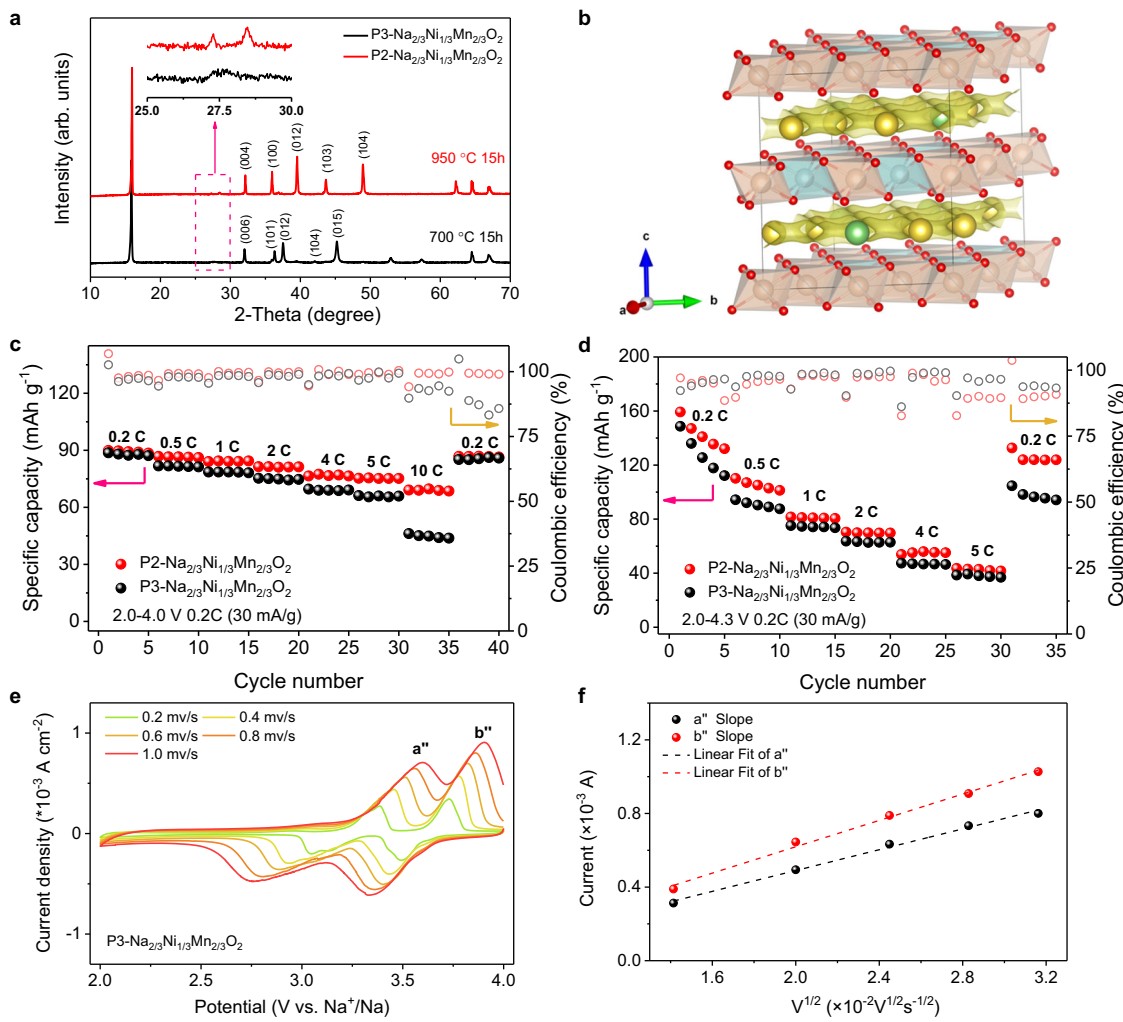

**Fig. 3 | Comparisons of Na$^+$ diffusion kinetics in P2- and P3-Na$_{2/3}$Ni$_{1/3}$Mn$_{2/3}$O$_2$ polymorphs. a** High-energy XRD patterns for Na$_{2/3}$Ni$_{1/3}$Mn$_{2/3}$O$_2$ precursor sintered at different temperatures. **b** Na$^+$ diffusion trajectories in P2-Na$_{2/3}$Ni$_{1/3}$Mn$_{2/3}$O$_2$ from BVEL mappings. The green ball represents Na$_f$ and the yellow ball stands for Na$_e$.

Rate performance of P2- and P3-Na$_{2/3}$Ni$_{1/3}$Mn$_{2/3}$O$_2$ cathodes between **c** 2.0–4.0 V and **d** 2.0–4.3 V. **e** CV curves of P3-Na$_{2/3}$Ni$_{1/3}$Mn$_{2/3}$O$_2$ electrode. **f** Dependence of peak current on the square root of scan rate ($v^{1/2}$) for P3-Na$_{2/3}$Ni$_{1/3}$Mn$_{2/3}$O$_2$ electrode.

aforementioned Na$^+$ LZZ ordering is energetically unfavorable within the P3-type layered oxides, which provides an ideal opportunity to elucidate the relationship between Na$^+$ ordering and diffusion kinetics. To this end, the P2- and P3- Na$_{2/3}$Ni$_{1/3}$Mn$_{2/3}$O$_2$ layered oxides are selected with the same composition hereafter[16,17,21]. As shown in Fig. 3a, the phase-pure P2-Na$_{2/3}$Ni$_{1/3}$Mn$_{2/3}$O$_2$ (space group: $P6_3$/$mmc$, No. 194) and P3-Na$_{2/3}$Ni$_{1/3}$Mn$_{2/3}$O$_2$ (space group: $R3m$, No. 160) materials are successfully prepared via a facile solid-state reaction at 950 and 700 °C, respectively. The SEM images indicate that the particle size of the P2 phase is much bigger than that of the P3 phase as shown in Fig. S14. From the XRD patterns, the presence of superstructure peaks in P2, but absence in P3 phases implies the Na$^+$ LZZ ordering within the Na slabs in P2 phase, which is consistent with the previous literatures[9,17] and the XRD results as well. While, the Ni−Mn intralayer ordering is well kept as expected in both P2- and P3-phases from the reported NPD data[16,17]. Under this circumstance, the BVEL calculations for the P2-Na$_{2/3}$Ni$_{1/3}$Mn$_{2/3}$O$_2$ (Fig. 3b) reveal a possible Na$^+$ diffusion pathway with an intermediate Na position at the adjacent sites, as is a similar case in P3 phase. Nevertheless, the structural discrepancies from the different lattice oxygen stacking cannot be ignored, including the complex charge ordering, the interactions among Na$^+$ and the repulsion between Na$^+$ and TM ions. To understand the underlying differences, the rate capability is firstly visited in a voltage range of 2–4.0 V for both

P2- and P3-Na$_{2/3}$Ni$_{1/3}$Mn$_{2/3}$O$_2$ half cells as shown in Fig. 3c. Note that there are no obvious phase transitions at this voltage range[21], and the intrinsic electrochemical kinetics difference between the two phases can be obtained without any external inferences (charge/discharge curves see Fig. S15). The P2-Na$_{2/3}$Ni$_{1/3}$Mn$_{2/3}$O$_2$ delivers 89.9 mA h g$^{-1}$ at 0.2 C (30 mA g$^{-1}$), 81.4 mA h g$^{-1}$ at 2.0 C, and 69.1 mA h g$^{-1}$ at 10.0 C, comparable with the results by Liu et al.[22] and Wu et al.[28]. In contrast, the rate performance of P3-Na$_{2/3}$Ni$_{1/3}$Mn$_{2/3}$O$_2$ is obviously lower than that of the P2 phase, especially at high rates, as is a similar case in P2- and P3-Na$_{0.62}$Ti$_{0.37}$Cr$_{0.63}$O$_2$ anodes[29]. Furthermore, although the continuous phase transition process occurs at a higher voltage (e.g., the P2-O2[21,30] and P3-O3[31] transitions), the overall rate performance of P2-Na$_{2/3}$Ni$_{1/3}$Mn$_{2/3}$O$_2$ is still obviously better than that of P3 one at a voltage range of 2.0−4.3 V as shown in Fig. 3d. To better understand this difference, Fig. 3e, f and S16 demonstrate the rate-varied CV curves and fitting results of both P2 and P3 electrodes. The two rate-dependent anodic CV peaks (a″ and b″) can be employed to analyze the Na$^+$ extraction kinetics in the P2- and P3-Na$_{2/3}$Ni$_{1/3}$Mn$_{2/3}$O$_2$ frameworks. The average slope of P2-Na$_{2/3}$Ni$_{1/3}$Mn$_{2/3}$O$_2$ is ~1.7 times larger than that of the P3 phase, which means the Na$^+$ diffusion coefficient in P2-phase is significantly higher than that in P3 one, perfectly accounting for the above rate performance differences. While, such a basic issue has not been fully understood ever before[32]. Moreover, the determined

apparent diffusion coefficients of Na$^+$ further confirm the above discussion as shown in Fig. S16 and Table S6. Of particular interest is that the essential crystal difference is the spatial distribution of Na prismatic sites induced by the stacking sequence of TMO$_6$ sheets in P2 and P3 phases. Hence, a concluding remark is that the different Na$^+$ mobilities are substantially resulted from the specific Na-ion/vacancy orderings in P2- and P3-Na$_{2/3}$Ni$_{1/3}$Mn$_{2/3}$O$_2$ electrodes. Therefore, the Na LZZ ordering should be responsible for the excellent rate capability in P2 phase. As a matter of fact, the present consensus is apt to support that the Na$^+$ diffusion in the O-type layered cathodes is harder than it in P-type ones due to the different coordination environment of Na$^+$ in O type (octahedral) and P-type (prismatic, the larger open space for Na$^+$ fast diffusion) materials[33,34]. The results here further point out the significant influence of the Na$^+$ occupancy induced by a slight compositional variation on the ionic migration within the P-type layered cathodes.

## Quantifying diffusional properties

Then, the AIMD and CINEB simulations are carefully conducted to explore the Na$^+$ diffusion in both P2 and P3 phases. Taking the different lattice oxygen stacking sequences into consideration, the site stability of Na$^+$ is calculated for the two prismatic sites, that is, the Na$_f$ (2b) site face sharing with the TMO$_6$ octahedra, and the Na$_e$ (2d) site edge sharing with the adjacent TMO$_6$ octahedra in P2-type layered oxides as shown in Fig. 4a, b. Note that the practical simulation can be simplified to calculate the total energies of Na$_e$ and Na$_f$ sites in a simple P2-Na$_x$CoO$_2$ model to comparably understand the corresponding stability. The calculated total energies revealed that the Na$_e$ site locates at a lower energy site of −168 meV/f. u. with respect to the Na$_f$ site in P2-NaCoO$_2$ as shown in Fig. 4c. Further structural enumeration in P2-Na$_{2/3}$CoO$_2$ indicate three types, viz., the honeycomb, chain and LZZ type Na ion orderings in P2 phase. Figure 4d indicates that the formation energy of the intralayer Na$^+$ LZZ ordering is lower than the Na$^+$ honeycomb and chain orderings for approx. 152 and 100 meV/f.u. in P2-Na$_{2/3}$Ni$_{1/3}$Mn$_{2/3}$O$_2$, respectively. Then, the DOSs of P2-Na$_{2/3}$Ni$_{1/3}$Mn$_{2/3}$O$_2$ with three different in-plane Na$^+$/vacancy orderings are calculated as shown in Fig. S17. The P2-Na$_{2/3}$Ni$_{1/3}$Mn$_{2/3}$O$_2$ with LZZ ordering exhibits an energy difference of 1.53 eV between the valence and conduction bands which is smaller than the other two orderings of 0.2 eV, which indicates the differences by the spatial distribution of Na$^+$. From a structural point of view, the structural stability of the P2 phase is not varying monotonously with the increased amount of Na$^+$ at Na$_e$ sites due to the Coulombic repulsion. Some Na$^+$ will alternatively occupy the Na$_f$ sites to balance and stabilize the complex interactions among the intralayer Na$^+$ in the Na slabs, and the strong correlations between the NaO$_6$ and TMO$_6$ polyhedra[35]. In contrast, in P3-Na$_{2/3}$Ni$_{1/3}$Mn$_{2/3}$O$_2$ phase, all the Na 3a sites are face and edge-sharing with the adjacent TMO$_6$ octahedra as equivalent crystal sites with an equal formation energy in an asymmetric environment along c-axis as shown in Fig. 4a, b.

The P2-Na$_{2/3}$Ni$_{1/3}$Mn$_{2/3}$O$_2$ with the disordered Na configurations (close to P2-Na$_{2/3}$Ni$_{0.3}$Mn$_{0.7}$O$_2$ as discussed above) were then enumerated computationally to examine the effect of Na$^+$ ordering on facilitating its diffusion (Supplementary Note 4). Figure 4e shows the total mean square displacements (MSDs) for Na-ions in P2- Na$_{2/3}$Ni$_{1/3}$Mn$_{2/3}$O$_2$ with Na LZZ ordering and disordering at 800 K. It can be seen that the MSDs of Na ion LZZ ordering is almost twice of it in disordering occupation, which means the enhanced Na-ion diffusivity after the structural disorder-to-order transition within the Na slabs. However, due to the limited computational resources and the large disordered supercell, it is difficult to take all the available diffusion paths into account for the detailed CI-NEB analysis. Considering that the Na$^+$ disordering occupation can be decomposed into a random combination of all the available Na$^+$ orderings, that is, the honeycomb, chain and LZZ types here, it emerges the chance to clarify the disorder-

to-order transition to accelerate the Na$^+$ diffusion in Na LZZ ordered structure. As a result, the MSDs of Na$^+$ by AIMD simulation in P2-Na$_{2/3}$Ni$_{1/3}$Mn$_{2/3}$O$_2$ with the all-possible ordered configurations at 800 K are presented in Fig. 4f. The noticeable migration of Na$^+$ is captured in the structure with the LZZ ordering, while no directional movement of Na$^+$ is found in the other two types of orderings. The probability density distribution (P(r)) of Na$^+$ at 800 K provides more intuitive results, where the diffusion pathways form a 2D network with which all of the Na$^+$ sites are connected via the adjacent vacancy as shown in Fig. 4i, to support the BVEL results as shown in Fig. 3b. In contrast, the Na$^+$ honeycomb and chain orderings are probably immobilized in the energetically favorable crystal sites with the negligible ionic motions, as is similar case in P2-Na$_{2/3}$CoO$_2$ as shown in Fig. S18.

To further disclose the underlying reasons, the CI-NEB calculations are performed in the ordered P2-Na$_{2/3}$Ni$_{1/3}$Mn$_{2/3}$O$_2$. Five Na ion diffusion trajectories are considered under monovacancy migration, including three paths of single ion migration (Na$_e$-Na$_e$ and Na$_f$-Na$_f$) and two paths of multiple Na$^+$ concerted migration (Paths 3 and 5), whose diffusion trajectories are illustrated in Fig. S19. The chain ordering structure is set as a reference due to its close formation energy with the LZZ structure and the higher occupancy of Na$_f$ sites. Figure 4j shows the activation energies ($E_{act}$) of Paths 1, 2, and 4 are obviously lower than those of the Na$^+$ concerted migration (respective ~0.29 and 0.44 eV for two Na$^+$ along Paths 3 and 5). For the single ion migration, the Na$^+$ along Path 2 (Na$_f$-Na$_f$) encounters a lowest $E_{act}$ of 0.09 eV, to imply the single migration dominated Na$^+$ diffusion within the chain ordering structure. Of special interest is that the single ion migration in LZZ ordering is not feasible since the Na$^+$ within the slab are highly correlated with each other. The movement of Na ion will inevitably lead to the involvement of Na ions nearby, which is actually the concerted migration of Na$^+$ as illustrated in Figs. S20, 4i–l. The $E_{act}$ of Na$_e$-Na$_e$ migration with the assistance of one Na$_f$ (Path 1) is estimated to be 0.05 eV, obviously lower than the Path 2 in chain ordering phase as shown in Fig. 4j. The $E_{act}$ of three Na$^+$ diffusion (Path 2) is also lower with regard to that of the single Na$^+$ migration. (Fig. 4k) Therefore, the fast-ionic diffusion kinetics in LZZ ordering phase can be ascribed to the multiple ion concerted migration that is originated from its special ionic occupations. A conclusion remark is that when the Na slab LZZ ordering is disrupted, but the TM honeycomb ordering is kept in Na$_{2/3}$Ni$_{0.3}$Mn$_{0.7}$O$_2$ material, the interaction between Na ions is weakened significantly (degenerates to a random combination of other types of Na$^+$ orderings) and the effectively concerted Na$^+$ diffusion can not be formed by the intrinsic structural constraints.

## AIMD simulations

Next, the underlying reasons for the high Na$^+$ diffusivity are understood in P2-Na$_{2/3}$Ni$_{1/3}$Mn$_{2/3}$O$_2$ using AIMD simulations. Note that the AIMD simulations collect the statistic contributions of all diffusional events and provide an effective complement to the CI-NEB method, especially for the complex coordination environment of the targeted materials[36]. Based on the structural enumeration[37], the supercells of P3-Na$_{2/3}$Ni$_{1/3}$Mn$_{2/3}$O$_2$ are successfully built as shown in Fig. S21, where there are two assembled types of Na$^+$ orderings in the supercell, and the energy differences of different Na arrangements are small enough (≤~20 meV/f. u.). This result indicates that from a thermodynamic point of view, the co-contribution from honeycomb and chain type Na occupations probably represents a none preferential in-plane Na ordering in P3- Na$_{2/3}$Ni$_{1/3}$Mn$_{2/3}$O$_2$ structure, in good agreement with the experimental results. However, the ground state structure is further simplified within a finite size supercell for the subsequent practical simulations. The temperature dependent MSDs of Na$^+$ in P2- and P3-Na$_{2/3}$Ni$_{1/3}$Mn$_{2/3}$O$_2$ (chain type) by AIMD simulations are shown in Fig. S22 as a function of time. The activation energy of Na$^+$ diffusion can be obtained according to the Arrhenius relation equation ($D = D_0 \exp\left(-\frac{E_a}{kT}\right)$) by fitting to the MSD results. The results indicate

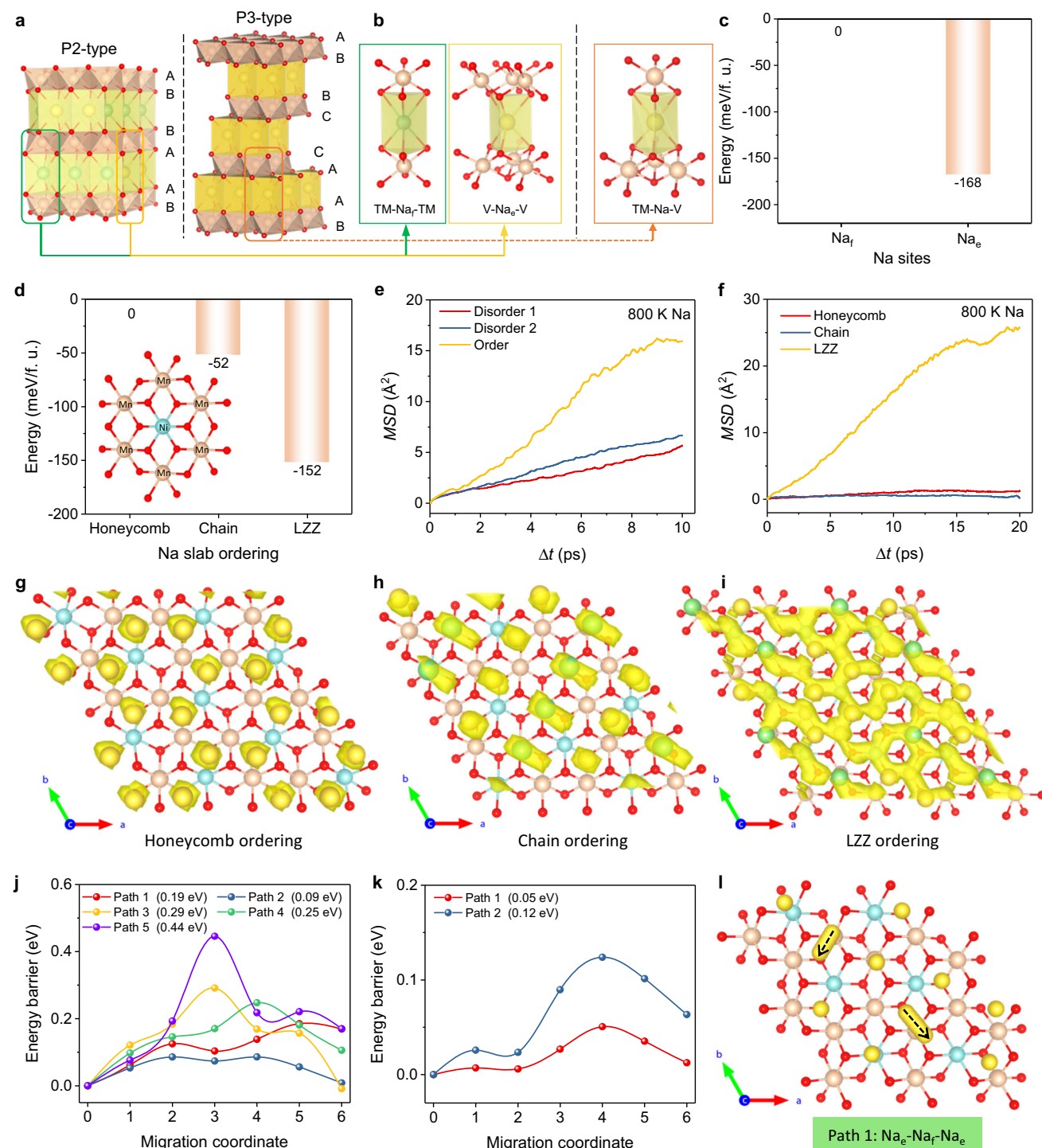

**Fig. 4 | Na$^+$ kinetics in P2/P3 Na$_{2/3}$Ni$_{1/3}$Mn$_{2/3}$O$_2$ layered oxides. a, b** Crystal environments of NaO$_6$ prism in both P2 and P3 phases, where the local coordination of the Na$_f$ (2b, faces-sharing) and Na$_e$ (2d, edge-sharing) sites in P2-phase and the Na (3a, face and edge-sharing) site in P3-phase are shown. **c** Calculated formation energies of Na$_e$ and Na$_f$ sites in a simple P2-NaCoO$_2$ model. **d** Energy differences of the in-plane Na$^+$ ordering in P2-Na$_{2/3}$Ni$_{1/3}$Mn$_{2/3}$O$_2$ structure, where the Mn$^{4+}$ and Ni$^{2+}$ ions described by a ($\sqrt{3} \times \sqrt{3}$)-R30° supercell in transition metal layer. **e** MSDs for Na-ions in P2- Na$_{2/3}$Ni$_{1/3}$Mn$_{2/3}$O$_2$ with Na$^+$ LZZ ordering and disordering at 800 K, including the Disorder 1, [occ. (Na$_f$) = 1/6] and Disorder 2, [occ. (Na$_f$) = 1/12]. **f** Total MSDs for Na-ions in P2-Na$_{2/3}$Ni$_{1/3}$Mn$_{2/3}$O$_2$ with different occupations of Na$^+$ at 800 K. Isosurface of the probability density distribution P(r) of Na$^+$ in P2-Na$_{2/3}$Ni$_{1/3}$Mn$_{2/3}$O$_2$ with **g** honeycomb, [occ. (Na$_f$) = 0] **h** chain, [occ. (Na$_f$) = 1/2] and **i** LZZ [occ. (Na$_f$) = 1/6] Na ion orderings within the Na slabs at 800 K, and the isosurface level is set to 0.001, where the green ball represents the Na$_f$ site, and the yellow one represents the Na$_e$ site. **j** Migration energy barriers for diffusion trajectories in layered P2-Na$_{2/3}$Ni$_{1/3}$Mn$_{2/3}$O$_2$ with chain and **k** LZZ orderings. **l** Illustration of Path 1 in layered P2-Na$_{2/3}$Ni$_{1/3}$Mn$_{2/3}$O$_2$ with LZZ ordering in **k**.

that with respect to the P3 structure (chain ordering), the P2 structure presents a higher diffusivity and lower activation energy of 200 meV as demonstrated in Fig. 5a, comparable with the results in P2-Na$_x$CoO$_2$ by Mo et al.[8]. In addition, the P3-Na$_{2/3}$Ni$_{1/3}$Mn$_{2/3}$O$_2$ (honeycomb ordering)

is also taken into considerations, and the AIMD result exhibits a much higher activation energy of 310 meV (refer to Fig. S25 and Supplementary Note 5). In summary, a much lower activation energy is observed in P2-Na$_{2/3}$Ni$_{1/3}$Mn$_{2/3}$O$_2$ with respect to that in both chain and

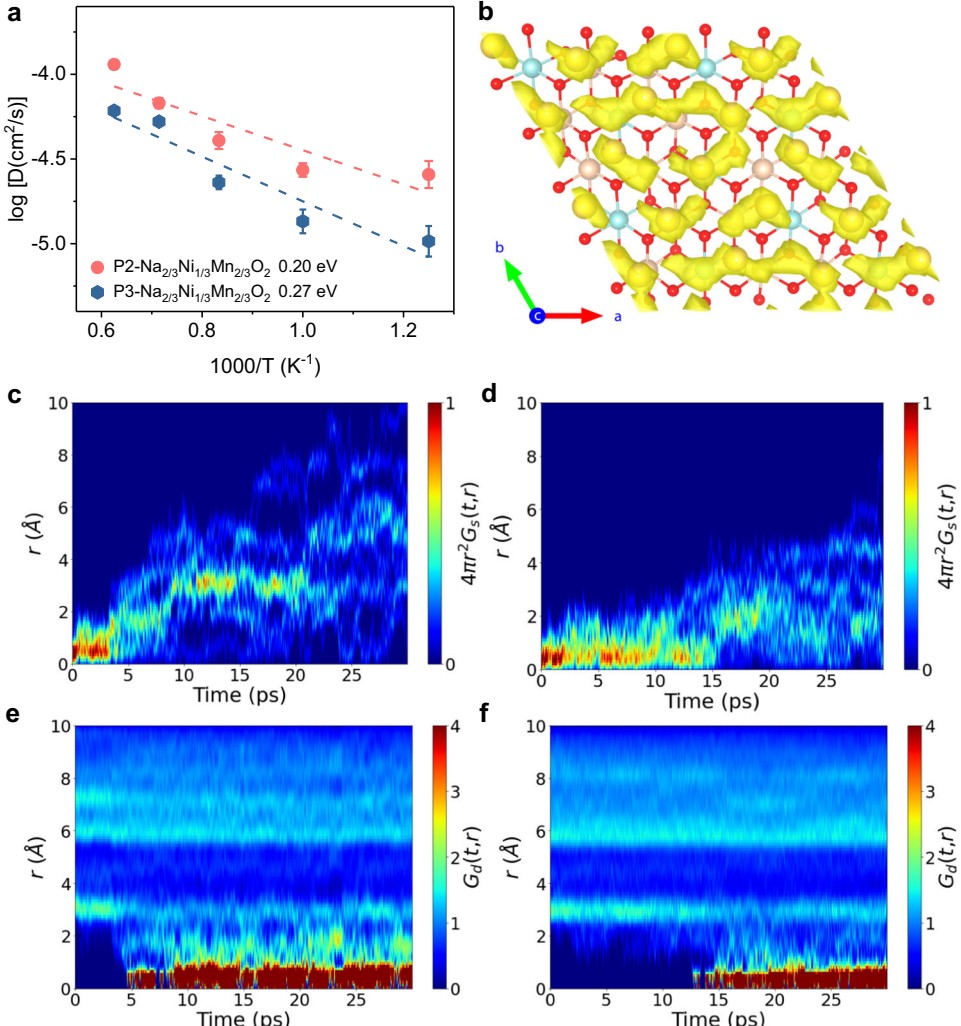

**Fig. 5 | Correlated sodium motions within Na slabs. a** Arrhenius plot of Na$^+$ diffusivity in P2 and P3-Na$_{2/3}$Ni$_{1/3}$Mn$_{2/3}$O$_2$ (chain ordering) from AIMD simulations. The error bars are the standard deviation of linear fit of MSD−Δ$t$ curves. The intralayer Na$^+$ diffusion pathway in **b** P3-Na$_{2/3}$Ni$_{1/3}$Mn$_{2/3}$O$_2$ from AIMD at 800 K. The self-part of the van Hove correlation function ($G_s$) for sodium in **c** P2- and **d** P3-Na$_{2/}$ $_3$Ni$_{1/3}$Mn$_{2/3}$O$_2$. The distinct part of the van Hove correlation function ($G_d$) for sodium ions in **e** P2- and **f** P3-Na$_{2/3}$Ni$_{1/3}$Mn$_{2/3}$O$_2$. Both $G_d$ and $G_s$ are functions of the average Na−Na pair distance ($r$) and time step after thermal equilibration at 800 K. The isosurface level of both is set to 0.001.

honeycomb ordering P3-Na$_{2/3}$Ni$_{1/3}$Mn$_{2/3}$O$_2$, which supports the above experimental results. Moreover, the experimental ultraviolet-visible absorption spectroscopy and the projected DOSs indicate that both P2- and P3-Na$_{2/3}$Ni$_{1/3}$Mn$_{2/3}$O$_2$ materials demonstrate a similar bandgap as shown in Figs. S23, S24, which probably implies the rate-determining step of the whole transport process does not lie in the electronic conductivity. To this end, the higher Na$^+$ ionic conductivity should be the main reason for the better rate performance of the P2-Na$_{2/3}$Ni$_{1/3}$Mn$_{2/3}$O$_2$ electrode as shown in Figs. 2d, 3c, d.

Then, the van Hove correlation function of Na$^+$ diffusion in P2 and P3-Na$_{2/3}$Ni$_{1/3}$Mn$_{2/3}$O$_2$ (chain type) is present in Fig. 5c–f using the ionic trajectories from AIMD simulations at 800 K, where the highly corre-lated ionic motions are usually investigated by the combination of the self-part ($G_s$) and the distinct part ($G_d$) of the van Hove correlation function ($G$)[38]. On one hand, the peak (red color) in $G_s$ between 0 and 2.0 Å links to the shortest Na−Na distances as shown in Fig. 5c. This distance increases rapidly after ~ 5 ps and then almost decays after ~20 ps, demonstrating that the Na$^+$ tend to leave their initial position to the following Na$^+$ sites in P2-Na$_{2/3}$Ni$_{1/3}$Mn$_{2/3}$O$_2$. In contrast, the peak in $G_s$ between 0 and 2.0 Å increases after ~15 ps, which is much longer than that of P2-Na$_{2/3}$Ni$_{1/3}$Mn$_{2/3}$O$_2$ as shown in Fig. 5d. It implied that the Na$^+$ have a higher probability of staying at the initial position and are

difficult to diffuse away to the neighboring sites in P3-Na$_{2/3}$Ni$_{1/3}$Mn$_{2/}$ $_3$O$_2$ (the AIMD simulation of P3-Na$_{2/3}$Ni$_{1/3}$Mn$_{2/3}$O$_2$ with honeycomb ordering is also provided in Fig. S25). On the other hand, in the van Hove correlation function, the radial distribution of the N−1 ions regarding the reference one after time $t$ is described by the distinct part $G_d(r, t)$[39], which could reveal the correlated Na$^+$ motions within the Na slabs in P2- and P3-Na$_{2/3}$Ni$_{1/3}$Mn$_{2/3}$O$_2$ as shown in Fig. 5e, f. The observed time scale of correlation in P2-Na$_{2/3}$Ni$_{1/3}$Mn$_{2/3}$O$_2$ is around 5 ps at 800 K, which is much shorter than that in P3-Na$_{2/3}$Ni$_{1/3}$Mn$_{2/3}$O$_2$ (-13 ps). Furthermore, the probability densities of Na$^+$ extracted from the trajectory at 800 K for P2-Na$_{2/3}$Ni$_{1/3}$Mn$_{2/3}$O$_2$ and P3-Na$_{2/3}$Ni$_{1/3}$Mn$_{2/}$ $_3$O$_2$ are shown in Figs. 4i, 5b, where the well-connected Na$^+$ diffusion channels are observed within the P2-Na$_{2/3}$Ni$_{1/3}$Mn$_{2/3}$O$_2$ framework. The higher energy Na$_f$ site also provides a medium for ion transport. While, in P3-Na$_{2/3}$Ni$_{1/3}$Mn$_{2/3}$O$_2$ structure, no connected channels are available within the Na slabs, and it seems difficult enough for one Na$^+$ diffusion to the adjacent sites as shown in Fig. 5b (Fig. S25).

These findings indicate that the Na$^+$ are taking a correlated dif-fusion way in P2-Na$_x$Ni$_{1/3}$Mn$_{2/3}$O$_2$ that embeds into the reconstruction of the new ordering structures upon desodiation. In other words, the Na$^+$ ordering is passively participating in the promotion of Na$^+$ fast diffusion upon cycling. The breaking and establishing of Na$^+$ ordering

themself acts fundamentally as the essential driving force to draw the P2-$Na_{2/3}Ni_{1/3}Mn_{2/3}O_2$ forth towards the high rate materials. Interpretations from the $^{23}Na$ solid-state nuclear magnetic resonance (ssNMR) on P2-$Na_{2/3}Ni_{1/3}Mn_{2/3}O_2$ by C. P. Gray et al.[40] also emphasized that the correlated in-plane $Na^+$ hopping between adjacent prismatic Na 2b and Na 2d sites contributes to the long-range $Na^+$ diffusion, here in consistence with the CI-NEB and AIMD results. After all, the significant structural difference is the Na ion occupation ways between the P2- and P3-$Na_{2/3}Ni_{1/3}Mn_{2/3}O_2$. In this regard, a small fraction of higher energy $Na_f$ occupation is a prerequisite for a lower migration barrier of $Na^+$ and probably also for the $Na^+$ orderings in P2-$Na_{2/3}Ni_{1/3}Mn_{2/3}O_2$. In other words, the above analysis explicitly points out the importance of the intralayer $Na^+$/vacancy ordering to the fast ionic migration.

## Discussion

In summary, a comprehensive study is performed to clarify the underlying relationship of $Na^+$/vacancy ordering on its diffusion kinetics in high energy density P-type layered oxides for SIBs. To verify this, two P2-type layered oxides are selected and the results demonstrate that the $Na^+$ diffusion coefficient is significantly reduced as the $Na^+$ LZZ ordering disappears gradually from the isostructural P2-$Na_{2/3}Ni_{1/3}Mn_{2/3}O_2$ to the P2-$Na_{2/3}Ni_{0.3}Mn_{0.7}O_2$ materials, although the transition metal ion ordering still presents. The CI-NEB simulation demonstrates that the concerted migration with a much lower energy barrier is captured in LZZ ordering concerning chain one. By introducing the different occupation ways of sodium ions between the P2- and P3-$Na_{2/3}Ni_{1/3}Mn_{2/3}O_2$, it is found that the LZZ ordering enables much higher $Na^+$ mobility, where the P2 structure has higher diffusivities and lower activation energy of 200 meV with respect to that in the P3 structure. The van Hove correlation function calculations demonstrate that the $Na^+$ in P2-$Na_{2/3}Ni_{1/3}Mn_{2/3}O_2$ has a much higher probability of taking a correlated diffusion way because a small fraction of $Na^+$ occupy the high energy sites in an inter-correlated way, which lead to the decrease of the energy barrier for the concerted $Na^+$ migration.

All these findings validate that the $Na^+$/vacancy ordering demonstrates a strong positive correlation with the fast $Na^+$ migration in P2 type layered cathodes. The ordering and disordering competition of corresponding ions ($Na^+$ and/or TM ions), especially in the spatial occupation, should motivate the fast ionic diffusion to fulfill the high-rate performance of rechargeable batteries. The attempts to establish the relationship between the $Na^+$ LZZ ordering and fast ionic kinetics in layered P2-type structures provide a rewarding avenue for designing high-rate electrode materials.

## Methods

### Sample preparation

The P2-$Na_{2/3}Ni_{1/3}Mn_{2/3}O_2$, P3-$Na_{2/3}Ni_{1/3}Mn_{2/3}O_2$, and P2-$Na_{2/3}Ni_{0.3}Mn_{0.7}O_2$ samples were synthesized via a facile solid-state reaction method using the stoichiometric amounts of $Na_2CO_3$ (99.99%, Macklin; an excessive 2% of $Na_2CO_3$ is added owing to the volatilization loss), $Mn_2O_3$ (99%, innochem), NiO (99.5%, Aladdin) as precursors. The mixed powders were ball milled for 24 h. Then, the powders were preheated in a muffle furnace at 450 °C for 6 h followed by annealing at different temperatures for 15 h. The products were cooled naturally down to room temperature, then ground and transferred to an argon-filled glovebox for protection.

### Characterizations

The X-ray diffraction (XRD) patterns were collected on a PANalytical Empyrean X-Ray Diffractometer (45 kV/50 mA) using a Cu $K_\alpha$ radiation ($\lambda \sim 1.5418$ Å) with the continuous scanning mode in a $2\theta$ range from 10° to 120°. The neutron diffraction was measured at the Multiple Physics Instrument (MPI) at the China Spallation Neutron Source (The wavelength range is 0.1 to 3 Å.). Roughly 2.0 g of sample was measured

for 6 h at ambient conditions (25 °C and 1 atm). The data reduction and correction were performed in the Mantid program[41]. The scattering data was corrected for absorption, but not for the multiple scattering (as the sample pathway is less than 1 cm). The MPI was equipped with 5 Banks for the data acquisition, from Bank 3 to 7. The Bank 6 data was selected for refinement because its data resolution is relatively high and the corresponding $d$-spacing covers most major peaks in the investigated structure, and its covers the d-spacing between 0.4 and 2.4 Å. According to the XRD and NPD simulations, the d-spacing of the superlattice peaks are located in the range of Bank 4 data, so the Bank 4 data is presented to demonstrate the structural difference of the as-prepared P2-$Na_{2/3}Ni_{1/3}Mn_{2/3}O_2$ and P2-$Na_{2/3}Ni_{0.3}Mn_{0.7}O_2$ samples. The GSAS II software was used to refine the X-ray and neutron data[42]. The scanning electron microscopy (SEM) and transmission electron microscopy (TEM) were carried out with a TESCAN MIRA4 (Czech Republic) and Talos F200X G2 TEM (ThermoFisher, American), respectively. The scanning transmission electron microscopy (STEM) was performed using a double-corrected FEI Titan G3 Cubed Themis 60–300 kV instrument operated at 200 kV. High-angle annular-dark-field (HAADF) STEM images were acquired with a convergence angle of 21 mrad and a HAADF inner–outer acceptance angles of 55–220 mrad. The STEM images were processed using a Wiener-filter (HREM Research and Gatan Digital Micrograph software). The ultraviolet-visible (UV) light absorption spectrum was recorded on the Perkin-Elmer Lambda 900 UV–VIS-NIR spectrometer to obtain the optic bandgap of the powder samples. The Raman spectra were acquired using inVia™ confocal Raman microscope (633 nm argon ion laser, Renishaw). Every spectrum was recorded with an exposure time of 20 s and accumulations of 3 times.

### Electrochemical measurements

The working cathode electrodes were prepared by mixing 80% active materials, 10% carbon black, 10% polyvinylidene fluoride (PVDF) binder and N-Methyl-2-pyrrolidone (NMP) together to form a slurry onto aluminum foil, then dried in a vacuum oven at 100 °C overnight. Then, the electrode sheets were cut into circular pieces with a diameter of 1.2 centimeters for coin-cell testing, and the mass loading of active material on Al was controlled at 1.8–2.0 mg cm$^{-2}$. The metallic sodium with a diameter of 1.4 centimeters and thickness of ~250 μm was used as the reference electrode with the 1 M $NaPF_6$ in propylene carbonate (PC) with 2 vol% FEC as the electrolyte for 100 μL and the glass fiber with diameter of 1.6 centimeters (Whatman, UK) as the separator. All the cells were assembled using CR2032 half coin cells in an Ar-filled MIKROUNA glovebox ($O_2$ and $H_2O < 0.1$ ppm). The electrochemical performance was tested using the Land CT2001A battery test system (Wuhan, China) in the voltage range of 2.0–4.0 V and 2.0–4.3 V. Note that the mass loading was kept the same during the electrochemical test for accurate comparison on kinetic properties. The cyclic voltammetry (CV) was performed at different scan rates of 0.2, 0.4, 0.6, 0.8, and 1.0 mV s$^{-1}$. The above-mentioned electrochemical measurements were conducted at 27 °C unless otherwise noted.

### Computational details

**DFT methods.** The total energy calculations were performed within the Vienna ab initio Simulation Package (VASP)[43,44] based on the DFT. The projected augmented wave (PAW)[45] potentials were used to deal with the electronic exchange-correlation interaction along with GGA functional in the parameterization of Perdew, Burke, and Ernzerhof (PBE) pseudopotential[46]. A plane wave representation for the wave function with a cut off energy of 500 eV was applied. The geometry optimizations were performed using a conjugated gradient minimization until all the forces acting on ions were less than 0.01 eV/Å per atom. We added a D3 correction to account for the Van der Waals interactions[47,48]. A $\Gamma$-centered k-point mesh of $3 \times 3 \times 2$ was used for the Brillouin zone samplings. The Hubbard parameter U for the GGA + U

calculations was 6.0 eV, 3.9 eV for Ni and Mn ions, respectively[49]. All calculations were performed with the default magnetic moments at the ground states. The crystal structures were built using VESTA software[50]. To find the ground-state structure, approx. 300 Na$^+$/vacancy structural orderings with different $Na_f$ ratios were enumerated[51] at the beginning. Then, the corresponding formation energies were calculated to set the lowest one as the ground state in P2-$Na_{2/3}CoO_2$, as is a similar case in P3-$Na_{2/3}CoO_2$. By fixing the arrangement of Na slab in the ground state structure, the Ni and Mn occupations were then generated with the honeycomb ordering in the transition metal layer that was derived from the NPD analysis. Afterwards, the above optimized crystal structures were transplanted to the P2- and P3-$Na_{2/3}Ni_{1/3}Mn_{2/3}O_2$ structures using the Ewald energy that is calculated in the Python Materials Genomics library[37] for further optimization. Finally, the one with the lowest total energy was set to the thermodynamically stable configurations of the corresponding P2 or P3 types. The $[2\sqrt{3a} \times 2\sqrt{3b} \times 1c]$-$R30°$ type supercell containing 88 atoms for P2-type $Na_{16}Ni_8Mn_{16}O_{48}$ and the $3a \times 3b \times 1c$ supercell includes 99 atoms for P3-type $Na_{18}Ni_9Mn_{18}O_{54}$ were adopted for geometry optimizations and kinetics simulations. Moreover, the $[4\sqrt{3a} \times 4\sqrt{3b} \times 1c]$-$R30°$-type superlattice including 352 atoms was built to simulate the disordered Na$^+$ occupations in P2-$Na_{2/3}Ni_{1/3}Mn_{2/3}O_2$. On the basis of the XRD/NPD results, the structure with [occ. ($Na_f$) = 1/12] and [occ. ($Na_f$) = 1/6] was setted to further obtain the two disordered Na configurations using Pymatgen and Enumlib[52]. The BVEL calculations were conducted to obtain the Na$^+$ migration pathways in layered oxides using the 3DBVSMAPPER code[53]. The Na ion migration barriers were calculated by climbing image nudged elastic band (CI-NEB) method[54,55] with 5 images as the intermediate states, where the CI-NEB simulation is considered to be complete when the magnitude of force per atom is smaller than 0.04 eV/Å.

**Ab initio molecular dynamics (AIMD) simulations.** The AIMD were carried out for the canonical (NVT) ensemble using a Nosé thermostat[56] at four elevated temperatures (800, 1000, 1200, 1400, and 1600 K). The volume and the shape of the cell were fixed. The corresponding structures were heated up to the targeted temperature by the velocity scaling over 3 ps, and then equilibrated at the desired temperature. The timescale of the simulations was 30 ps and a time step of 1 fs was used to integrate the equation of motion. Upon the data analysis, the dynamic process of the last 5 ps was not considered to improve the fitting results[36]. To save the computational resources reasonably, the integration in reciprocal space was performed only at the Γ-point.

The diffusion coefficient is calculated using the following formula,

$$D = \lim_{t \to \infty} \left[ \frac{1}{2dt} \left\langle \left[ \vec{r}_i(t) \right]^2 \right\rangle \right] \tag{1}$$

Where the $d$ is the dimension of the lattice in which the diffusion occurs, $t$ is the elapsed time. The average mean square displacement (MSD) is defined as

$$MSD(t) = \frac{1}{N} \sum_{i=1}^{N} \left\langle \left[ \vec{r}_i(t+t_0) \right]^2 - \left[ \vec{r}_i(t_0) \right]^2 \right\rangle \tag{2}$$

Where the $r_i(t)$ is the displacement of the $i$th Na$^+$ at the time $t$, and the $t_0$ is the starting time. The diffusion coefficient $D$ is obtained by linearly fitting to the dependence of average MSD over $2dt$. The diffusivity in certain direction is obtained by fitting to the MSD in these directions over time. The probability density of mobile ion was obtained by ensemble averaging over the trajectories[39].

**Correlation of ion dynamics.** The van Hove correlation function analysis[38,57] was conducted to comprehend the correlations during the

Na ionic motions, where the self-part $G_s$ and the distinct-part $G_d$ of the van Hove correlation function ($G$) can be expressed as follows.

$$G_s(r,t) = \frac{1}{4\pi r^2 N_d} \left\langle \sum_{i=1}^{N_d} \delta\left(r - |\mathbf{r}_i(t_0) - \mathbf{r}_i(t+t_0)|\right) \right\rangle_{t_0} \tag{3}$$

$$G_d(r,t) = \frac{1}{4\pi r^2 \rho N_d} \left\langle \sum_{i \neq j}^{N_d} \delta\left(r - |\mathbf{r}_i(t_0) - \mathbf{r}_j(t+t_0)|\right) \right\rangle_{t_0} \tag{4}$$

Here, the $\delta(\cdot)$ represents the one-dimensional Dirac delta function. The angular bracket is the ensemble average over the initial time $t_0$. The $\mathbf{r}_i(t)$ stands for the position of the $i$th ions at the time $t$. The $N_d$ and r are the diffusing sodium ions in the unit cell and the radial distance, respectively. The $\rho$ is the average number density which serves as the "normalization factor" in $G_d$.

## Data availability
The authors declare that all data supporting the finding of this study are available within the paper and its supplementary information files. All raw data generated during the current study are available from the corresponding authors upon request.

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

## Acknowledgements
This work was supported by National Key Research and Development Project (2019YFA0705702), National Natural Science Foundation of China (22075328, 22109186), Guangdong Basic and Applied Basic Research Foundation (2021B1515120002), Hundreds of Talents program of Sun Yat-sen University and the Natural Science Foundation of Guangdong Province (2022A1515010405). Computational resources were provided by the Tianhe-2 National Supercomputer Center in Guangzhou. We appreciate the neutron beamtime granted from MPI in China Spallation Neutron Source.

## Author contributions
Y.S.S. and X.L. conceived the project. W.H.K. collected and analyzed the neutron powder diffraction. Y.S.S. performed the materials synthesis and electrochemical testing. Y.S.S., P.F.J., S.C.W., B.W., X.Y.L., and G.Y.Q. performed the characterizations. Y.S.S., W.X.C., and Y.S. performed the computational calculations. Y.S.S., Y.S., and X.L. analyzed the data and proposed the mechanisms. Y.S.S. wrote the draft that was revised and finalized by X.L. All authors discussed the results and commented on the manuscript.

## Competing interests
The authors declare no competing interests.
