## [Peer Review File · Nature Communications]

Reviewer comments, first round

Reviewer #1 (Remarks to the Author):

This is an interesting piece of work on the understanding fast Na⁺ ionic diffusion in P2-type Na₂/3Ni₁/3Mn₂/3O₂. Though there are some useful insights, particularly on the theory side. I found the authors need to better present their structure part. It is also quite possible that there is monoclinic distortion in the system, I would suggest the authors to index their neutron diffraction pattern, and carry out more rigorous structural analysis, some trial space groups are P21/m or P21/c for their P2 phase. Some more detailed comments/suggestions:

Page 2 line 41 the authors stated "... as well as the far less than known Na diffusion kinetics", this is not true. Na diffusion are broadly investigated, isotope diffusion experiment was even carried out for single crystal beta-alumina...

Page 2 line 49 the authors stated "... The symbols of "P" or "O" refer to the spatial arrangements of oxygen ions, where the "P" corresponds to the prismatic sublattice and the "O" is the octahedral sublattice in the layered structure ". This is the incorrect interpretation or description of the "P" or "O" type structure. The P and O are not used to describe the spatial arrangements of the oxygen ions, though they do have some connections to some extent, but not necessarily directly correlated. "P" and "O" are used to describe the coordination environments of interstitial cations, e.g., Na⁺ or Li⁺, whether they are prismatically or octahedrally coordinated to the neighboring oxygen ions. The spatial arrangements of oxygen ions are described by the stacking sequences of the anion, such as AB... or ABC... or AABB... etc.

Glad to see data from MPI instrument from CSNS, but the description of data collection, reduction and the way of data analysis is handled are quite disappointing. First, MPI is a new neutron total scattering instrument, the authors (especially the beamline scientist) have the responsibility to establish a more rigorous way in reducing and modeling the diffraction data. The extremely simple description "The neutron diffraction data was measured at the Multiple Physics Instrument (MPI) at China Spallation Neutron Source. Roughly 2 g of sample was measured for 6 h at ambient condition. The GSAS II software was used to refine the X-ray and Neutron data." is too vague for any serious structure studies, which seems to be an important part of this study. My suggestions for the authors, please add the following information: " what is the wavelength range used? What is ambient condition? What routine was used to reduce the data? Were the data corrected for absorption/multiple scattering? What does bank 4 mean in the data, what angle range does bank 4 cover? Why the authors show bank 4 in the main text, is it the highest resolution bank? Why there is no neutron PDF data, since that is quite important for confirming short-range cation ordering..."

Since the authors have already calculated the Van Hove correlation function, if they use the calculated $G(r,t)$ around $t = 0$, that should lead to the energy non-discriminated pair correlation function, or instantaneous PDF. Since neutron total scattering data were collected from MPI, why don't the authors compare the theoretically calculated total PDF to the experimental neutron PDF data?

Though there is likely some level of inelastic distortion for the experimental neutron total scattering data, but that shall not prevent the qualitative comparison. It is a pity that no neutron PDF data are used (data were collected from a dedicated neutron total scattering instrument!). Neutron PDF should also be very useful in understanding the short-range ordering of both Ni/Mn and Na⁺/vacancy. My speculation is that monoclinic shearing is often associated with the zig-zag type Na⁺-vacancy ordering in this type of system. It has been previously observed in both P2 and P3 Na-ion systems with honeycomb ordered cations on the transition metal layer. This is also not surprise to me since the zig-zag ordered Na⁺-vacancy configuration has the lowest electrostatic repulsion from the neighboring TM layer.

Overall, I think a major revision is needed before I could suggest for further consideration in Nature Communications.

Reviewer #2 (Remarks to the Author):

The manuscript tends to prove that the slightly compositional change and the structural alternation

can modify the Na ion/vacancy ordering, and therefore regulates the Na⁺ diffusion through the concerted migration. Experimental characterizations, which focus on structural information and electrochemical performance, and DFT calculations were combined to validate the manuscript's statement. Overall, while the work seems interest to me and can supplement the literature, the currently insufficient data and the major concerns to prove the statement of the manuscripts urge me fail to recommend the publication of the paper at the current stage, pending on the revision from the authors.

Major concerns:

(1) The authors used only one additional composition (i.e., Na₂/3Ni_{0.3}Mn_{0.7}O₂) to illustrate the deviation from the optimal Na₂/3Ni₁/3Mn₂/3O₂ composition can modulate the Na ordering and the electrochemical performance, which is not convincing and may result from the coincidence. More compositions (at least one more) deviated from the optimal Na₂/3Ni₁/3Mn₂/3O₂ should be discussed in the manuscript. After adding more compositions and their performances, the effect of the composition on the Na configurations and the electrochemical performance can be clearly drawn.

(2) In Figure 1, the explanation of using the XRD and the NPD pattern to prove the existence of Na LZZ ordering and the TM honeycomb ordering is poorly demonstrated, especially for the NPD data. I even start to understand how XRD works after reading over multiple references, which should have been clearly explained in this paper. Thus, the authors should clearly demonstrate the correlation between minor peak changes (peaks in XRD and NPD) and the corresponding structural configurations. Why do those minor changes in the XRD and NPD pattern can indicate the ordering of Na and TM.

(3) In Figure 2 cd, the SAED pattern of disorder Na and TM is lacked as referenced samples to finally prove the existence of Na and TM order. The authors ought to provide simulated disorder SEAD images to make comparison.

(4) The usage of the P3 structure to prove the the effect of Na⁺ ordering on facilitating Na⁺ diffusion seems weird for me. Why did not the author directly utilize the P2 structure with disorder Na configurations to support their statement, instead of the involvement of new P3 structure with new influential factors? Authors are suggested to discuss the electrochemical performance and the diffusion patterns in disorder Na₂/3Ni₁/3Mn₂/3O₂ and make comparison with the order one. If it is difficult to synthesize the disorder P2-Na₂/3Ni₁/3Mn₂/3O₂ in experiments, computational approach is a good choice.

(5) I question the method in the paper that using only 20 candidate structures with the lowest Ewald energies can find the appropriate ground-state structures with the Na LZZ and TM honeycomb order. In contrast to the manuscripts, Meng et. al. (J. Chem. Phys. 2008, 128, 104708) used the complicated cluster expansion method to identify the Na LZZ order, and One et. al. (Phys. Rev. APPLIED 7, 064003 2017) enumerated 5000 structures to finally found the ground state structures. The authors should disclose more details of their methods for finding ground state structures.

(6) The P3 structure used in the computations is inconsistent with the experimental characterizations. Since the authors declare that Na is disorder in the P3-Na₂/3Ni₁/3Mn₂/3O₂, why did the authors used the P3 structure with order honeycomb Na (Figure S14) for computations? The authors are suggested to clarify the choice carefully, because the wrong use of the ground state structure may disqualify the whole computations.

(7) The direct and convincing evidence of the concerted migration to facilitate the diffusion is to perform the NEB calculations in both single and concerted ways, and compare their migration energies, as demonstrated in multiple studies (e.g., Nature communications 8, 15893, 2017 and Solid State Ionics, 371, 115767, 2021). The currently usage of the Van Hoff function to prove the concerted migration is far from enough to support the concerted migration mechanism, and it is deemed as a side evidence. Thus, the authors ought to perform the NEB calculations to prove the concerted migration can lower down the migration energies as illustrated in their table of content. The NEB migration energies can also be compared with those in P2-Na₂/3Ni_{0.3}Mn_{0.7}O₂, disorder P2-Na₂/3Ni₁/3Mn₂/3O₂ (suggested in comment 4) and P3-Na₂/3Ni₁/3Mn₂/3O₂ to finally prove the facilitation of Na ordering on diffusion.

Minor comments:

(1) The equations used in the paper are often lacked for description of each factors, e.g., line 149 ρ and line 226.

(2) The description in page 3 "the influence of the Na⁺-ion/vacancy ordering on Na kinetics for P2-type layered oxides has been less visited because of the significant challenges of finding a suitable material system to configure its contribution out solely.

" may not be fair. A simple search can reveal related studies, e.g., Chemistry of Materials 26 (18), 5208-5214, 2014.

(3) The authors should justify the usage of the Hubbard U potential in their calculations. Their U parameters' choice is even different from the their reference (Ref. 25). Authors are encouraged to calibrate their U parameters by aligning them to their experimental bandgap.

(4) The authors should clarify their statement in page 9, i.e., Of particular interest is that the IE_{2g}/IA_{1g} ratio indicates the spatial distribution and occupation of Na⁺ ions to some extent. It is unacceptable to just provide the conclusion without any explanation or references.

(5) In page 13, the authors concluded that electronic structures remains unchanged in P2-Na₂/3Ni_{0.3}Mn_{0.7}O₂ and P2-Na₂/3Ni₁/3Mn₂/3O₂ from the bandgap characterization, which is not convincing. The authors are suggested to perform the DFT calculations of the band structures to support their statement.

(6) The content from line 295 to 300 is too puzzled to understand, please rephrase the sentences.

(7) In Figure 4i, the authors are suggested to change the scale of the y axis from the Ln based on e to log based on 10 to increase the readability as previous computational papers.

(8) In Figure 4i, the authors are suggested to add the error bar of each calculated diffusivity following the established method.

REVIEWER COMMENTS

Reviewer #1 (Remarks to the Author):

This is an interesting piece of work on the understanding fast Na⁺ ionic diffusion in P2-type Na_{2/3}Ni_{1/3}Mn_{2/3}O₂. Though there are some useful insights, particularly on the theory side. I found the authors need to better present their structure part.

It is also quite possible that there is monoclinic distortion in the system, I would suggest the authors to index their neutron diffraction pattern, and carry out more rigorous structural analysis, some trial space groups are $P2_1/m$ or $P2_1/c$ for their P2 phase.

Reply:

To identify the phase/atomic structures of the as-prepared materials, we have performed the XRD, NPD, Raman spectra and electron microscopy characterizations that are consistent with each other to reveal the structural occupation of the Na⁺ ion on its transport in the P type layered cathodes. Furthermore, systematic XRD and NPD simulations are applied to understand the relationship between the diffraction pattern and the underlying atomic arrangement. Moreover, the XRD simulation on four different space groups of monoclinic distortion is provided in Figure R1. The diffraction peaks of the monoclinic phase in Figure R1b are absent in experimental data in Figure 1a of the revised manuscript (peaks at 27.3° and 28.4°). We did not find evident monoclinic distortion upon the XRD analysis. Then, we considered the proposed space groups of $P2_1/m$, $C2/c$, $C2/m$ or $P2_1/c$ to perform the refinements of the experimental XRD and NPD, the fitting results of which seem not supportive enough to the above ones. Hence, we believe that the presence of the monoclinic phase can be ruled out in the P2-Na_{2/3}Ni_{1/3}Mn_{2/3}O₂ and P2-Na_{2/3}Ni_{0.3}Mn_{0.7}O₂ material. And the extra diffraction peaks can be ascribed to the superlattice ordering of transition metal ions and Na⁺ ions in P2 type oxides, which will be further discussed below.

Figure R1. (a) Simulated XRD patterns of $\text{Na}_x\text{Co}_y\text{O}_z$ (these structures can be found in open-access database (<https://materialsproject.org/#>) according to the materials id.) with four different space groups of monoclinic distortion. (b) A magnified view of XRD patterns between 20-38°

In detail, three types of in-plane Na^+ -ion/vacancy orderings can be found in the P2 phase according to the structural enumeration upon DFT simulations. Figure R2 (Figure S2 in SI) shows the simulated XRD patterns of the above ordering structures using RIETAN-FP. Owing to the similar X-ray scattering factor of Ni and Mn ions, they can not be distinguished clearly. Therefore, the small superlattice peaks in X-ray diffraction patterns can be assigned to the in-plane Na^+ -ion/vacancy orderings, corresponding to the d-spacings of the adjacent intralayer sodium ions. In Figure 1 of the revised manuscript, the clear LZZ superstructure peaks are found at 27.3° and 28.4° for $\text{P2-Na}_{2/3}\text{Ni}_{1/3}\text{Mn}_{2/3}\text{O}_2$ (inset of Figure 1a), corresponding to the d-spacings of 3.13 and 3.26 Å, respectively, which are absent in the $\text{Na}_{2/3}\text{Ni}_{0.3}\text{Mn}_{0.7}\text{O}_2$ pattern to show the in-plane Na^+ -ion/vacancy differences. Hence, this also indicates the presence of the LZZ ordering in $\text{P2-Na}_{2/3}\text{Ni}_{1/3}\text{Mn}_{2/3}\text{O}_2$ by XRD refinement, as shown in Figure R3 (Figure S1 in SI).

Figure R2 (Figure S2 in SI). (a) Simulated XRD patterns of P2-type layered oxides with three different in-plane Na⁺-ion/vacancy ordering (large zigzag, honeycomb and chain type). (b) A magnified view of XRD patterns between 20-38°, where the d-spacings correspond to the average distances between the adjacent intralayer sodium ions in the in-plane Na⁺-ion/vacancy ordering arrangements. The atomic structures of P2-Na_{2/3}Ni_{1/3}Mn_{2/3}O₂ with three different in-plane Na⁺-ion/vacancy orderings, (c) honeycomb type, [occ (Na_f) = 0]; (d) chain type, [occ (Na_f) = 1/2] and (e) large zigzag (LZZ), [occ (Na_f) = 1/6]. The green ball represents the Na_f site and the yellow ball stands for Na_e site. (f) The atomic arrangements of Mn⁴⁺ and Ni²⁺ ions described by honeycomb ordering in the transition metal planes of P2-Na_{2/3}Ni_{1/3}Mn_{2/3}O₂.

Figure R3 (Figure S1 a in SI). Refined XRD pattern for P2-Na_{2/3}Ni_{1/3}Mn_{2/3}O₂.

As for the neutron powder diffraction (NPD), the strong neutron scattering length contrasts of Ni (10.3 fm) and Mn (-3.73 fm) enable it to be an effective measurement to detect the Ni–Mn intralayer ordering as shown in Figure 1b of the revised manuscript. Figure R4 (Figure S3 in SI) shows the peak differences of transition metal (TM) honeycomb ordering and Na slab large zigzag (LZZ) ordering in the NPD results. The *d*-spacings of 3.13 and 3.26 Å indicate the Na slab LZZ ordering in the red shadowed regions, which are in line with the experimental data in Figure 1b of the revised manuscript. Moreover, the simulation results point out that the *d*-spacings of 3.42, 4.02 and 3.26 Å can be assigned to the TM honeycomb ordering, which are shadowed with the yellow color in Figure 1b and Figure R4 (Figure S3 in SI).

Figure R4 (Figure S3 in SI). Simulated NPD data of P2-type layered oxides with the transition metal (TM) honeycomb ordering, the Na slab large zigzag(LZZ) ordering and the coexistence of TM honeycomb and Na LZZ ordering using GSAS II.

Hence, the structural orderings in $P2\text{-Na}_{2/3}\text{Ni}_{1/3}\text{Mn}_{2/3}\text{O}_2$ can be undoubtedly identified using XRD and NPD characterizations, where those minor peaks and the corresponding atomic structures can be clearly figured out in the following DFT simulations. These ordering arrangements of Na, Ni and Mn, we think, contribute to the structural stability (less distortions), and the transitions from the disorder to order accelerate the Na^+ ion diffusion kinetics. The following electrochemistry, kinetic analysis and the competitions among different ordering phases are all derived.

To improve the readability, discussions are supplemented on Pages 4-5 of the revised supporting information as follows.

“The XRD/NPD data analysis is based on crystallography, where the phenomenal peaks are the direct responses to the spatial ordering arrangement of one or more species of atoms in one specific crystal

structure. In terms of superstructures, they always respond with small intensities at lower diffraction angles, where they have larger lattice constants than the original structure. Hence, the high energy XRD, or NPD, is generally required to characterize these small or minor peaks.

In detail, three types of in-plane Na⁺-ion/vacancy orderings can be found in P2 phase according to the structural enumeration upon DFT simulations. Figure S2 shows the simulated XRD patterns of the above ordering structures using RIETAN-FP. Owing to the similar X-ray scattering factor of Ni and Mn ions, they can not be distinguished clearly. While, the small superlattice peaks in X-ray diffraction patterns can be assigned to the in-plane Na⁺-ion/vacancy orderings, corresponding to the d-spacings of the adjacent intralayer sodium ions. In Figure 1 of the manuscript, the clear LZZ superstructure peaks are found at 27.3° and 28.4° for P2-Na_{2/3}Ni_{1/3}Mn_{2/3}O₂ (inset of Figure 1a), corresponding to the d-spacings of 3.13 and 3.26 Å, respectively, which are absent in the Na_{2/3}Ni_{0.3}Mn_{0.7}O₂ pattern to show the structural differences.”

Some more detailed comments/suggestions:

Page 2 line 41 the authors stated “... as well as the far less than known Na diffusion kinetics”, this is not true. Na diffusion are broadly investigated, isotope diffusion experiment was even carried out for single crystal beta-alumina...

Reply:

Firstly, we agree that the Na⁺ ion kinetics are broadly investigated in the single crystal beta-alumina, especially in the superior ionic conductivity phase, where the high Na⁺ ion diffusion coefficient can be measured.

While, when it comes to the concerted Na⁺ ion diffusion mechanism, it seems at present we can not get a full physical picture to describe the Na⁺ ion diffusion behavior, as is a similar case in the layered cathodes materials to transport the Li⁺ or K⁺ ions (this is what we want to state in the manuscript).

Our previous results see references below.

[1] Gao, A.; Xia, J.; Li, M.; Lu, X.; Wang, F.; Yang, R., Water–Solid Interface Engineering Stabilizes K-Birnessite Cathode. Adv. Funct. Mater. 2021, 32, 2108267.

[2] Gao, A.; Sun, Y.; Zhang, Q. H.; Zheng, J. Y.; Lu, X., Evolution of Ni/Li Antisites under the Phase

Transition of a Layered LiNi_{1/3}Co_{1/3}Mn_{1/3}O₂ Cathode. J. Mater. Chem. A 2020, 8, 6337-6348.

[3] Gao, A.; Li, M.; Guo, N. N.; Qiu, D. P.; Li, Y.; Wang, S. H.; Lu, X.; Wang, F.; Yang, R., K-Birnessite Electrode Obtained by Ion Exchange for Potassium-Ion Batteries: Insight into the Concerted Ionic Diffusion and K Storage Mechanism. Adv. Energy Mater. 2019, 9, 1802739.

[4] Lu, X.; Wang, S.; Xiao, R.; Shi, S.; Li, H.; Chen, L., First-Principles Insight into the Structural Fundamental of Super Ionic Conducting in NASICON MTi₂(PO₄)₃ (M = Li, Na) Materials for Rechargeable Batteries. Nano Energy 2017, 41, 626-633.

[5] Sun, Y.; Lu, X.; Xiao, R. J.; Li, H.; Huang, X. J., Kinetically Controlled Lithium-Staging in Delithiated LiFePO₄ Driven by the Fe Center Mediated Interlayer Li-Li Interactions. Chem. Mat. 2012, 24, 4693-4703.

To make this sentence more accurate, it is modified as follows.

In Page 2 of the revised manuscript:

“From a material’s perspective of view, the problem lies mainly in electrode materials, including the lack of alternative candidates and the ambiguous mechanism between Na diffusion kinetics and atomic structure upon battery cycling.”

Page 2 line 49 the authors stated “ ... The symbols of “P” or “O” refer to the spatial arrangements of oxygen ions, where the “P” corresponds to the prismatic sublattice and the “O” is the octahedral sublattice in the layered structure ”. This is the incorrect interpretation or description of the “P” or “O” type structure. The P and O are not used to describe the spatial arrangements of the oxygen ions, though they do have some connections to some extent, but not necessarily directly correlated. “P” and “O” are used to describe the coordination environments of interstitial cations, e.g., Na⁺ or Li⁺, whether they are prismatically or octahedrally coordinated to the neighboring oxygen ions. The spatial arrangements of oxygen ions are described by the stacking sequences of the anion, such as AB... or ABC... or AABB... etc.

Reply:

We agree with that the “P” and “O” are used to describe the coordination environments of interstitial

cations, e.g., Na⁺ or Li⁺, whether they are prismatically or octahedrally coordinated to the neighboring oxygen ions, *although they do have some connections to the spatial arrangements of the oxygen ions as denoted by Demas et. al., (Physica B+C 99.1-4 (1980): 81-85.)*

In Page 2 of the revised manuscript, we have supplemented the discussions to assure the exact expression of local structural features as follows.

“The symbols of “P” or “O” refer to the coordination environments of interstitial cations, e.g., Na⁺ or Li⁺, where they are prismatically or octahedrally coordinated to the neighboring oxygen ions.”

Glad to see data from MPI instrument from CSNS, but the description of data collection, reduction and the way of data analysis is handled are quite disappointing. First, MPI is a new neutron total scattering instrument, the authors (especially the beamline scientist) have the responsibility to establish a more rigorous way in reducing and modeling the diffraction data. The extremely simple description “The neutron diffraction data was measured at the Multiple Physics Instrument (MPI) at China Spallation Neutron Source. Roughly 2 g of sample was measured for 6 h at ambient condition. The GSAS II software was used to refine the X-ray and Neutron data.” is too vague for any serious structure studies, which seems to be an important part of this study. My suggestions for the authors, please add the following information: “what is the wavelength range used? What is ambient condition? What routine was used to reduce the data? Were the data corrected for absorption/multiple scattering? What does bank 4 mean in the data, what angle range does bank 4 cover? Why the authors show bank 4 in the main text, is it the highest resolution bank? Why there is no neutron PDF data, since that is quite important for confirming short-range cation ordering...”

Reply:

Thank you for your questioning these important issues.

Now, our collaborators at the China Spallation Neutron Source provide the corresponding information as follows for your reference.

The wavelength range was 0.1 to 3 Å. The samples were measured at ambient conditions (25 °C and 1 atm). The data reduction and correction were performed in the Mantid program. The scattering data

corrected for absorption, but not the multiple scattering (as the sample pathway is less than 1 cm). The MPI was equipped with 5 banks for the data acquisition, denoted as bank 3 to bank 7. The Bank 6 data was used for the refinement study, which covers the d-spacing between 0.4 Å and 2.4 Å. We selected bank 6 because the data resolution is relatively high and the corresponding d-spacing covers most major peaks in the investigated structure. The Bank 7 is the highest resolution bank but its d-spacing range is very limited. The d-spacing of bank 4 data (1.0 Å to 4.5 Å) covers angle range of 20-100° (steady state (CW) Neutron wavelength: 1.6215 Å) in NPD measurements.

According to the XRD and NPD simulations, the d-spacing of the superlattice peaks are located in the range of the Bank 4 data, so the Bank 4 data is presented in the main text to demonstrate the structural difference of the as-prepared P2-Na_{2/3}Ni_{1/3}Mn_{2/3}O₂ and P2-Na_{2/3}Ni_{0.3}Mn_{0.7}O₂ samples.

Since the data acquisition time and the weight of samples were limited, the current data quality was not good enough for PDF analysis.

The related discussions have been supplemented in Page 4-5 of revised manuscript.

“The neutron diffraction was measured at the Multiple Physics Instrument (MPI) at the China Spallation Neutron Source (The wavelength range is 0.1 to 3 Å.). Roughly 2.0 g of sample was measured for 6 h at ambient conditions (25 °C and 1 atm). The data reduction and correction were performed in the Mantid program.¹⁸ The scattering data was corrected for absorption, but not for the multiple scattering (as the sample pathway is less than 1 cm). The MPI was equipped with 5 Banks for the data acquisition, from Bank 3 to 7. The Bank 6 data was selected for refinement because its data resolution is relatively high and the corresponding d-spacing covers most major peaks in the investigated structure, and its covers the d-spacing between 0.4 and 2.4 Å. According to the XRD and NPD simulations, the d-spacing of the superlattice peaks are located in the range of Bank 4 data, so the Bank 4 data is presented to demonstrate the structural difference of the as-prepared P2-Na_{2/3}Ni_{1/3}Mn_{2/3}O₂ and P2-Na_{2/3}Ni_{0.3}Mn_{0.7}O₂ samples. The GSAS II software was used to refine the X-ray and neutron data.¹⁹”

Since the authors have already calculated the Van Hove correlation function, if they use the calculated $G(r, t)$ around $t = 0$, that should lead to the energy non-discriminated pair correlation function, or

instantons PDF. Since neutron total scattering data were collected from MPI, why don't the authors compare the theoretically calculated total PDF to the experiment neutron PDF data? Though there is likely some level of inelastic distortion for the experimental neutron total scattering data, but that shall not prevent the qualitative comparison. It is a pity that no neutron PDF data are used (data were collected from a dedicated neutron total scattering instrument!). Neutron PDF should also be very useful in understanding the short-range ordering of both Ni/Mn and Na⁺/vacancy. My speculation is that monoclinic sheering is often associated with the zig-zag type Na⁺-vacancy ordering in this type of system. It has been previously observed in both P2 and P3 Na-ion systems with honeycomb ordered cations on the transition metal layer. This is also not surprise to me since the zig-zag ordered Na⁺-vacancy configuration has the lowest electrostatic repulsion from the neighboring TM layer.

Reply:

First of all, we thank for your expertise and discussion to the NPD/neutron PDF and the structural understandings to layered cathode materials. We basically agree with your speculation "My speculation is that monoclinic sheering is often associated with the zig-zag type Na⁺-vacancy ordering in this type of system. It has been previously observed in both P2 and P3 Na-ion systems with honeycomb ordered cations on the transition metal layer. This is also not surprise to me since the zig-zag ordered Na⁺-vacancy configuration has the lowest electrostatic repulsion from the neighboring TM layer." The high-quality PDF data can truly perform as a useful method to investigate the structural orderings in layered cathodes, but we are not able to get high-quality PDF data at present.

But even with the PDF data, this is still a challenging work to be established with significant efforts, because the structural ordering in layered cathode for battery application stays mysterious, where the spatial ordering in this layered phase varies with the dopant content and crystal site, etc.. Further, there are significant uncertainties in these ordering structures which change with the Na content, the dopant amount, the current rate, the electric field applied and even the temperature (Li₂FeSiO₄ system, J. Power Sources 2016, 318, 136-145), etc.. The perfect understanding of the underlying structural evolution seems difficult at present, especially in a background of a non-equilibrated electrochemical cycling process. However, we will continue to strive to get more interesting results.

Then, we greatly appreciate your comments and help to decode the structural ordering in this layered

cathode to optimize this work. We also feel that “it is a pity that no neutron PDF data are collected...”, which is limited by the real test conditions. De facto, the qualitative comparison between the theoretically calculated total PDF to the experiment neutron PDF data approach is a feasible way to understand the short-range ordering of both Ni/Mn and Na⁺/vacancy in NaNiMnO₂ based layered cathodes, which has been considered in previous literature (e.g., Chem. Mater. 2016, 28, 6817–6821). Specifically, the Neutron PDF data were collected to investigate the Ni/Mn ordering in lithium nickel manganese oxide (Li-Ni_{0.5}Mn_{1.5}O₄) spinel to understand the structure–electrochemical performance. However, the above experiment was conducted on materials with a strong neutron scattering contrast. While, it is a little different here, where the slight compositional variation induced structural disorder-to-order of Na⁺ ion is our target. And the long range of Ni/Mn orderings remains unchanged in P2-Na_{2/3}Ni_{1/3}Mn_{2/3}O₂ and P2-Na_{2/3}Ni_{0.3}Mn_{0.7}O₂ electrodes as depicted in NPD data. Of particular importance is that the intensity of superstructure peaks induced by the Na⁺ ions LZZ ordering are too small to investigate with respect to the Ni/Mn ordering (Refer to Figure 1b). To further figure out the underlying fast Na⁺ ion migration mechanism, we supplement more detailed experimental and theoretical results to analyze the role of LZZ ordering in P2-Na_{2/3}Ni_{1/3}Mn_{2/3}O₂.

Discussion has been added in Pages 19-22 of the revised manuscript as follows.

“Then, the AIMD and CINEB simulations are carefully conducted to explore the Na⁺ ion diffusion in both P2 and P3 phases. Taking the different lattice oxygen stacking sequences into consideration, the site stability of Na⁺ ion is calculated for the two prismatic sites, that is, the Na_f (2b) site face sharing with the TMO₆ octahedra, and the Na_e (2d) site edge sharing with the adjacent TMO₆ octahedra in P2-type layered oxides as shown in Figure 4a, b. Note that the practical simulation can be simplified to calculate the total energies of Na_e and Na_f sites in a simple P2-Na_xCoO₂ model to comparably understand the corresponding stability. The calculated total energies revealed that the Na_e site locates at a lower energy site of -168 meV/f. u. with respect to the Na_f site in P2-NaCoO₂ as shown in Figure 4c. Further structural enumeration in P2-Na_{2/3}CoO₂ indicate three types, viz., the honeycomb, chain and LZZ type Na ion orderings in P2 phase. Figure 4d indicates that the formation energy of the intralayer Na⁺ ion LZZ ordering is lower than the Na⁺ ion honeycomb and chain orderings for approx. 152 and 100 meV/f.u. in P2-Na_{2/3}Ni_{1/3}Mn_{2/3}O₂, respectively. Then, the DOSs of P2-Na_{2/3}Ni_{1/3}Mn_{2/3}O₂

with three different in-plane Na⁺-ion/vacancy orderings are calculated as shown in Figure S17. The P2-Na_{2/3}Ni_{1/3}Mn_{2/3}O₂ with LZZ ordering exhibits an energy difference of 1.53 eV between the valence and conduction bands which is smaller than the other two orderings of 0.2 eV, which indicates the differences by the spatial distribution of Na⁺ ions. From a structural point of view, the structural stability of the P2 phase is not varying monotonously with the increased amount of Na⁺ ions at Na_e sites due to the Coulombic repulsion. Some Na⁺ ions will alternatively occupy the Na_f sites to balance and stabilize the complex interactions among the intralayer Na⁺ ions in the Na slabs, and the strong correlations between the NaO₆ and TMO₆ polyhedra.⁵⁶ In contrast, in P3-Na_{2/3}Ni_{1/3}Mn_{2/3}O₂ phase, all the Na 3a sites are face and edge-sharing with the adjacent TMO₆ octahedra as equivalent crystal sites with an equal formation energy in an asymmetric environment along c axis as shown in Figure 4a, b. The P2-Na_{2/3}Ni_{1/3}Mn_{2/3}O₂ with the disordered Na configurations (close to P2-Na_{2/3}Ni_{0.3}Mn_{0.7}O₂ as discussed above) were then enumerated computationally to examine the effect of Na⁺ ion ordering on facilitating its diffusion (Supplementary Note 4). Figure 4e shows the total mean square displacements (MSDs) for Na-ions in P2- Na_{2/3}Ni_{1/3}Mn_{2/3}O₂ with Na LZZ ordering and disordering at 800 K. It can be seen that the MSDs of Na ion LZZ ordering is almost twice of it in disordering occupation, which means the enhanced Na-ion diffusivity after the structural disorder-to-order transition within the Na slabs. However, due to the limited computational resources and the large disordered supercell, it is difficult to take all the available diffusion paths into account for the detailed CI-NEB analysis. Considering that the Na⁺ ion disordering occupation can be decomposed into a random combination of all the available Na⁺ ion orderings, that is, the honeycomb, chain and LZZ types here, it emerges the chance to clarify the disorder-to-order transition to accelerate the Na⁺ ion diffusion in Na LZZ ordered structure. As a result, the MSDs of Na⁺ ions by AIMD simulation in P2-Na_{2/3}Ni_{1/3}Mn_{2/3}O₂ with the all-possible ordered configurations at 800 K are presented in Figure 4f. The noticeable migration of Na⁺ ions is captured in the structure with the LZZ ordering, while no directional movement of Na⁺ ions is found in the other two types of orderings. The probability density distribution (P(r)) of Na⁺ ions at 800 K provides more intuitive results, where the diffusion pathways form a 2D network with which all of the Na⁺ ion sites are connected via the adjacent vacancy as shown in Figure 4i, to support the BVEL results as shown in Figure 3b. In contrast, the Na⁺ ion honeycomb and chain orderings are

probably immobilized in the energetically favorable crystal sites with the negligible ionic motions, as is similar case in P2- $\text{Na}_{2/3}\text{CoO}_2$ as shown in Figure S18.

To further disclose the underlying reasons, the CI-NEB calculations are performed in the ordered P2- $\text{Na}_{2/3}\text{Ni}_{1/3}\text{Mn}_{2/3}\text{O}_2$. Five Na ion diffusion trajectories are considered under monovacancy migration, including three paths of single ion migration ($\text{Na}_e\text{-Na}_e$ and $\text{Na}_f\text{-Na}_f$) and two paths of multiple Na^+ ion concerted migration (Paths 3 and 5), whose diffusion trajectories are illustrated in Figure S19. The chain ordering structure is set as a reference due to its close formation energy with the LZZ structure and the higher occupancy of Na_f sites. Figure 4j shows the activation energies (E_{act}) of Paths 1, 2 and 4 are obviously lower than those of the Na^+ ions concerted migration (respective ~ 0.29 and 0.44 eV for two Na^+ ion along Paths 3 and 5). For the single ion migration, the Na^+ ion along Path 2 ($\text{Na}_f\text{-Na}_f$) encounters a lowest E_{act} of 0.09 eV, to imply the single migration dominated Na^+ ion diffusion within the chain ordering structure. Of special interest is that the single ion migration in LZZ ordering is not feasible since the Na^+ ions within the slab are highly correlated with each other. The movement of Na ion will inevitably lead to the involvement of Na ions nearby, which is actually the concerted migration of Na^+ ions as illustrated in Figures S20 and 4i-l. The E_{act} of $\text{Na}_e\text{-Na}_e$ migration with the assistance of one Na_f (Path 1) is estimated to be 0.05 eV, obviously lower than the Path 2 in chain ordering phase as shown in Figure 4j. The E_{act} of three Na^+ ion diffusion (Path 2) is also lower with regard to that of the single Na^+ ion migration. (Figure 4k) Therefore, the fast-ionic diffusion kinetics in LZZ ordering phase can be ascribed to the multiple ion concerted migration that is originated from its special ionic occupations. A conclusion remark is that when the Na slab LZZ ordering is disrupted, but the TM honeycomb ordering is kept in $\text{Na}_{2/3}\text{Ni}_{0.3}\text{Mn}_{0.7}\text{O}_2$ material, the interaction between Na ions is weakened significantly (degenerates to a random combination of other types of Na^+ orderings) and the effectively concerted Na^+ ion diffusion can not be formed by the intrinsic structural constraints.”

Figure 4. Na^+ ion kinetics in P2/P3 $\text{Na}_{2/3}\text{Ni}_{1/3}\text{Mn}_{2/3}\text{O}_2$ layered oxides. (a, b) Crystal environments of NaO_6 prism in both P2 and P3 phases, where the local coordination of the Na_f (2b, faces-sharing) and Na_e (2d, edge-sharing) sites in P2-phase and the Na (3a, face and edge-sharing) site in P3-phase are shown. (c) Calculated formation energies of Na_e and Na_f sites in a simple P2- NaCoO_2 model. (d) Energy differences of the in-plane Na^+ ion ordering in P2- $\text{Na}_{2/3}\text{Ni}_{1/3}\text{Mn}_{2/3}\text{O}_2$ structure, where the Mn^{4+} and Ni^{2+} ions described by a $(\sqrt{3} \times \sqrt{3})\text{-R}30^\circ$ supercell in transition metal layer. (e) MSDs for Na-ions in P2- $\text{Na}_{2/3}\text{Ni}_{1/3}\text{Mn}_{2/3}\text{O}_2$ with Na^+ ion LZZ ordering and disordering at 800 K, including the Disorder 1, [occ. (Na_f) = 1/6] and Disorder 2, [occ. (Na_f) = 1/12]. (f) Total MSDs for Na-ions in P2- $\text{Na}_{2/3}\text{Ni}_{1/3}\text{Mn}_{2/3}\text{O}_2$ with different occupations of Na at 800K. Isosurface of the probability density distribution $P(r)$ of Na^+ in P2- $\text{Na}_{2/3}\text{Ni}_{1/3}\text{Mn}_{2/3}\text{O}_2$ with (g) honeycomb, [occ. (Na_f) = 0] (h) chain, [occ. (Na_f) = 1/2] and (i) LZZ [occ. (Na_f) = 1/6] Na ion orderings within the Na slabs at 800 K, and the isosurface level is set to 0.001, where the green ball represents the Na_f site, and the yellow one represents the Na_e site. (j) Migration energy barriers for diffusion trajectories in layered P2- $\text{Na}_{2/3}\text{Ni}_{1/3}\text{Mn}_{2/3}\text{O}_2$ with chain

and (k) LZZ orderings. (i) Illustration of Path 1 in layered P2-Na_{2/3}Ni_{1/3}Mn_{2/3}O₂ with LZZ ordering in Figure 4k.

Overall, I think a major revision is needed before I could suggest for further consideration in Nature Communications.

Thanks for your discussion and comments!

Reviewer #2 (Remarks to the Author):

The manuscript tends to prove that the slightly compositional change and the structural alternation can modify the Na ion/vacancy ordering, and therefore regulates the Na⁺ diffusion through the concerted migration. Experimental characterizations, which focus on structural information and electrochemical performance, and DFT calculations were combined to validate the manuscript's statement. Overall, while the work seems interest to me and can supplement the literature, the currently insufficient data and the major concerns to prove the statement of the manuscripts urge me fail to recommend the publication of the paper at the current stage, pending on the revision from the authors.

Major concerns:

(1) The authors used only one additional composition (i.e., Na_{2/3}Ni_{0.3}Mn_{0.7}O₂) to illustrate the deviation from the optimal Na_{2/3}Ni_{1/3}Mn_{2/3}O₂ composition can modulate the Na ordering and the electrochemical performance, which is not convincing and may result from the coincidence. More compositions (at least one more) deviated from the optimal Na_{2/3}Ni_{1/3}Mn_{2/3}O₂ should be discussed in the manuscript. After adding more compositions and their performances, the effect of the composition on the Na configurations and the electrochemical performance can be clearly drawn.

Reply:

In response to this concern, one more deviation of the P2-Na_{2/3}Ni_{1/3}Mn_{2/3}O₂ phase (Ni > 1/3 is thermodynamically unreachable in P2 type oxides), that is, the Al doped P2-Na_{2/3}Al_{1/24}Ni_{7/24}Mn_{2/3}O₂ (structural perturbation by Al doping), is prepared to further re-check the electrochemical performance difference that is relating with the Na ion occupations as shown in Figure R5 (Figure S11 in SI). It can be seen that a tiny composition modulation does not cause significant changes in the shape of the electrochemical curve with only the disruption of LZZ ordering at the Na slab. as is a similar case in the subsequent rate capability tests in P2-Na_{2/3}Ni_{0.3}Mn_{0.7}O₂ electrode (Figure S12 and Table S5). Furthermore, the superlattice peaks in XRD and NPD data nearly disappear in P2-Na_{2/3}[Co_xNi_{1/3-x}Mn_{2/3}]O₂ after introducing 1/6 Co, (Chem. Mater. 2000, 12, 3583-3590) also indicating that both of the Na ion ordering and the spatial distribution of transition metal cations are altered as well (see

XRD and Raman data in Figure S13). The electrochemical performance of P2- $\text{Na}_{2/3}\text{Co}_{1/6}\text{Ni}_{1/6}\text{Mn}_{2/3}\text{O}_2$ is also inferior to that of P2- $\text{Na}_{2/3}\text{Ni}_{1/3}\text{Mn}_{2/3}\text{O}_2$ at high current density. Hence, the fast Na^+ ion diffusion is strongly correlated with the Na^+ -ion/vacancy ordering and is probably accelerated by it.

Figure R5 (Figure S11 in SI). Charge/discharge curves of P2- $\text{Na}_{2/3}\text{Ni}_{1/3}\text{Mn}_{2/3}\text{O}_2$, P2- $\text{Na}_{2/3}\text{Ni}_{0.3}\text{Mn}_{0.7}\text{O}_2$ and P2- $\text{Na}_{2/3}\text{Al}_{1/24}\text{Ni}_{7/24}\text{Mn}_{2/3}\text{O}_2$ samples at 0.2 C (30 mA g^{-1}) in the 2nd cycle between 2.0 and 4.0 V.

Figure R6 (Figure S12 in SI). (a) XRD patterns of the as-prepared $P2\text{-Na}_{2/3}\text{Ni}_{1/3}\text{Mn}_{2/3}\text{O}_2$ and $P2\text{-Na}_{2/3}\text{Al}_{1/24}\text{Ni}_{7/24}\text{Mn}_{2/3}\text{O}_2$ sintered at 950°C for 15h, where the inset indicates the loss of in-plane Na^+ -ion/vacancy ordering in $P2\text{-Na}_{2/3}\text{Al}_{1/24}\text{Ni}_{7/24}\text{Mn}_{2/3}\text{O}_2$. (b) Raman spectra of $P2\text{-Na}_{2/3}\text{Ni}_{1/3}\text{Mn}_{2/3}\text{O}_2$ and $P2\text{-Na}_{2/3}\text{Al}_{1/24}\text{Ni}_{7/24}\text{Mn}_{2/3}\text{O}_2$. (c) Charge/discharge curves of $P2\text{-Na}_{2/3}\text{Al}_{1/24}\text{Ni}_{7/24}\text{Mn}_{2/3}\text{O}_2$ between 2.0 and 4.0 V at 0.2 C. (d) Rate performance. (e) CV curves of $P2\text{-Na}_{2/3}\text{Al}_{1/24}\text{Ni}_{7/24}\text{Mn}_{2/3}\text{O}_2$ electrode at different scanning rates between 2.0 - 4.0 V. (f) Dependence of peak currents as marked in Figure S12e on the square root of the scan

rates ($v^{1/2}$) for P2- $\text{Na}_{2/3}\text{Al}_{1/24}\text{Ni}_{7/24}\text{Mn}_{2/3}\text{O}_2$ electrode.

Figure R7 (Figure S13 in SI). (a) XRD patterns of the as-prepared P2- $\text{Na}_{2/3}\text{Ni}_{1/3}\text{Mn}_{2/3}\text{O}_2$ and P2- $\text{Na}_{2/3}\text{Co}_{1/6}\text{Ni}_{1/6}\text{Mn}_{2/3}\text{O}_2$ sintered at 950°C for 15h, where the inset indicates the loss of in-plane Na^+ -ion/vacancy ordering in P2- $\text{Na}_{2/3}\text{Co}_{1/6}\text{Ni}_{1/6}\text{Mn}_{2/3}\text{O}_2$. (b) Raman spectra of P2- $\text{Na}_{2/3}\text{Ni}_{1/3}\text{Mn}_{2/3}\text{O}_2$ and P2- $\text{Na}_{2/3}\text{Co}_{1/6}\text{Ni}_{1/6}\text{Mn}_{2/3}\text{O}_2$. (c) Charge/discharge curves of P2- $\text{Na}_{2/3}\text{Co}_{1/6}\text{Ni}_{1/6}\text{Mn}_{2/3}\text{O}_2$ between 2.0 and 4.0 V at 0.2 C. (d) Rate performance.

The related discussions have been added in Pages 14-15 of the revised manuscript.

“In addition, one more deviation of the P2- $\text{Na}_{2/3}\text{Ni}_{1/3}\text{Mn}_{2/3}\text{O}_2$ phase is the minor Al doped P2- $\text{Na}_{2/3}\text{Al}_{1/24}\text{Ni}_{7/24}\text{Mn}_{2/3}\text{O}_2$ (structural perturbation by Al doping) is prepared to re-check the electrochemical performance difference that relates with the Na ion occupations. This tiny composition modulation does not cause significant changes in the shape of the electrochemical curve, but disrupts the Na slab LZZ ordering as shown in Figure S12. The subsequent rate capability test shows similar

results as disclosed in the P2-Na_{2/3}Ni_{0.3}Mn_{0.7}O₂ (Figures S8, S9, S12 and Table S5). Furthermore, the absence of superlattice peaks in another composition as the P2-Na_{2/3}[Co_xNi_{1/3-x}Mn_{2/3}]O₂ by introducing 1/6 Co into the transition layers also means the disappearance of the ordered Na ion and transition metal cation arrangements as reported¹⁶ (see Figure S13, and supplementary Note 3). The electrochemical performance of P2-Na_{2/3}Co_{1/6}Ni_{1/6}Mn_{2/3}O₂ is also inferior to that of P2-Na_{2/3}Ni_{1/3}Mn_{2/3}O₂ at high current density. Hence, the fast Na⁺ ion diffusion is strongly correlated with the Na⁺-ion/vacancy ordering, and probably is accelerated by it.”

(2) In Figure 1, the explanation of using the XRD and the NPD pattern to prove the existence of Na LZZ ordering and the TM honeycomb ordering is poorly demonstrated, especially for the NPD data. I even start to understand how XRD works after reading over multiple references, which should have been clearly explained in this paper. Thus, the authors should clearly demonstrate the correlation between minor peak changes (peaks in XRD and NPD) and the corresponding structural configurations. Why do those minor changes in the XRD and NPD pattern can indicate the ordering of Na and TM.

Reply:

Thanks for your suggestion.

The XRD/NPD data analysis is based on crystallography, where the phenomenal peaks are the direct responses to the spatial ordering arrangement of one or more species of atoms in one specific crystal structure. In terms of superstructures, they always respond with small intensities at lower diffraction angles, where they have larger lattice constants than the original structure. To characterize these small or minor peaks, the high energy XRD, or NPD, is in general required upon acquiring the data process. Here, significant works are performed to identify the minor XRD/NPD peaks in P2-Na_{2/3}Ni_{1/3}Mn_{2/3}O₂ based materials, including the atomic structural analysis and artificial simulation of XRD/NPD patterns and the DFT theoretical calculation of the atomic occupations.

In details, three types of in-plane Na⁺-ion/vacancy orderings can be found in the P2 phase according to the structural enumeration upon DFT simulations. Figure R2 (Figure S2 in SI) shows the simulated XRD patterns of the above ordering structures using RIETAN-FP. Owing to the similar X-ray

scattering factor of Ni and Mn ions, they can not be distinguished clearly. While, the small superlattice peaks in X-ray diffraction patterns can be assigned to the in-plane Na^+ -ion/vacancy orderings, corresponding to the d -spacings of the adjacent intralayer sodium ions. In Figure 1a of the revised manuscript, the clear LZZ superstructure peaks are found at 27.3° and 28.4° for $\text{P2-Na}_{2/3}\text{Ni}_{1/3}\text{Mn}_{2/3}\text{O}_2$ (inset of Figure 1a), corresponding to the d -spacings of 3.13 and 3.26 Å, respectively, which are absent in the $\text{Na}_{2/3}\text{Ni}_{0.3}\text{Mn}_{0.7}\text{O}_2$ pattern to show the structural differences. Hence, this also indicates the presence of the LZZ ordering in $\text{P2-Na}_{2/3}\text{Ni}_{1/3}\text{Mn}_{2/3}\text{O}_2$ by XRD characterization as shown in Figure R3 (Figure S1 in SI).

Figure R2 (Figure S2 in SI). (a) Simulated XRD patterns of P2-type layered oxides with three different in-plane Na^+ -ion/vacancy ordering (large zigzag, honeycomb and chain type). (b) A magnified view of XRD

patterns between 20-38°, where the d -spacings correspond to the average distances between the adjacent intralayer sodium ions in the in-plane Na^+ -ion/vacancy ordering arrangements. The atomic structures of $\text{P2-Na}_{2/3}\text{Ni}_{1/3}\text{Mn}_{2/3}\text{O}_2$ with three different in-plane Na^+ -ion/vacancy orderings, (c) honeycomb type, [$\text{occ}(\text{Na}_f) = 0$]; (d) chain type, [$\text{occ}(\text{Na}_f) = 1/2$] and (e) large zigzag (LZZ), [$\text{occ}(\text{Na}_f) = 1/6$]. The green ball represents the Na_f site and the yellow ball stands for Na_e site. (f) The atomic arrangements of Mn^{4+} and Ni^{2+} ions described by honeycomb ordering in the transition metal planes of $\text{P2-Na}_{2/3}\text{Ni}_{1/3}\text{Mn}_{2/3}\text{O}_2$.

Figure R3 (Figure S1 a in SI). Refined XRD pattern for $\text{P2-Na}_{2/3}\text{Ni}_{1/3}\text{Mn}_{2/3}\text{O}_2$.

As for the neutron powder diffraction (NPD), the strong neutron scattering length contrasts of Ni (10.3 fm) and Mn (-3.73 fm) enable it to be an effective measurement to detect the Ni–Mn intralayer ordering as shown in Figure 1b of the revised manuscript. Figure R4 (Figure S3 in SI) shows the peak differences of transition metal (TM) honeycomb ordering and Na slab large zigzag (LZZ) ordering in the NPD results. The d -spacings of 3.13 and 3.26 Å indicate the Na slab LZZ) ordering in the red shadowed regions, which are in line with the experimental data in Figure 1b of the revised manuscript. Moreover, the simulation results point out that the d -spacings of 3.42, 4.02 and 3.26 Å can be assigned to the TM honeycomb ordering, which are shadowed with the yellow color in Figure 1b and Figure R4 (Figure S3 in SI).

Figure R4 (Figure S3 in SI). Simulated NPD data of P2-type layered oxides with the transition metal (TM)honeycomb ordering, the Na slab large zigzag(LZZ) ordering and the coexistence of TM honeycomb and Na LZZ ordering using GSAS II.

To make it easy to read, discussions are supplemented on Pages 4-5 of the revised supporting information (Supplementary Notes 1-2) as follows.

“The XRD/NPD data analysis is based on crystallography, where the phenomenal peaks are the direct responses to the spatial ordering arrangement of one or more species of atoms in one specific crystal structure. In terms of superstructures, they always respond with small intensities at lower diffraction angles, where they have larger lattice constants than the original structure. Hence, the high energy XRD, or NPD, is generally required to characterize these small or minor peaks.

In detail, three types of in-plane Na⁺-ion/vacancy orderings can be found in P2 phase according to the structural enumeration upon DFT simulations. Figure S2 shows the simulated XRD patterns of the above ordering structures using RIETAN-FP. Owing to the similar X-ray scattering factor of Ni and Mn ions, they can not be distinguished clearly. While, the small superlattice peaks in X-ray diffraction

patterns can be assigned to the in-plane Na⁺-ion/vacancy orderings, corresponding to the d-spacings of the adjacent intralayer sodium ions. In Figure 1 of the manuscript, the clear LZZ superstructure peaks are found at 27.3° and 28.4° for P2-Na_{2/3}Ni_{1/3}Mn_{2/3}O₂ (inset of Figure 1a), corresponding to the d-spacings of 3.13 and 3.26 Å, respectively, which are absent in the Na_{2/3}Ni_{0.3}Mn_{0.7}O₂ pattern to show the structural differences.

As for the neutron powder diffraction (NPD), the strong neutron scattering length contrasts of Ni (10.3 fm) and Mn (-3.73 fm) enable it to be an effective measurement to detect the Ni–Mn intralayer ordering as shown in Figure 1b. Figure S3 shows the NPD results' peak differences between transition metal (TM) honeycomb and Na slab large zigzag (LZZ) ordering. The d-spacings of 3.13 and 3.26 Å indicate the Na slab LZZ ordering in the red shadowed regions, which are in line with the experimental data in Figure 1b of the revised manuscript. Moreover, the simulation results point out that the d-spacings of 3.42, 4.02 and 3.26 Å can be assigned to the TM honeycomb ordering, which are shadowed with the yellow color in Figure 1b and Figure S3 in SI.”

(3) In Figure 2 c, d, the SAED pattern of disorder Na and TM is lacked as referenced samples to finally prove the existence of Na and TM order. The authors ought to provide simulated disorder SEAD images to make comparison.

Reply:

The simulated disorder SEAD image has been supplemented in Figure R8 (Figure S6 in SI). The extra diffraction spots induced by Na slab LZZ and TM honeycomb ordering are clearly shown.

Figure R8 (Figure S6 in SI). Structure of P2-type layered oxide. The Na slab in (a) LLZ ordering, (c) TM honeycomb ordering and (e) Na slab disordering with (b, d, f) respective simulated SAED patterns at [001] zone axis.

(4) The usage of the P3 structure to prove the effect of Na^+ ordering on facilitating Na^+ diffusion seems weird for me. Why did not the author directly utilize the P2 structure with disorder Na configurations to support their statement, instead of the involvement of new P3 structure with new influential factors? Authors are suggested to discuss the electrochemical performance and the diffusion patterns in disorder $\text{Na}_{2/3}\text{Ni}_{1/3}\text{Mn}_{2/3}\text{O}_2$ and make comparison with the order one. If it is difficult to synthesize the disorder P2- $\text{Na}_{2/3}\text{Ni}_{1/3}\text{Mn}_{2/3}\text{O}_2$ in experiments, computational approach is a good choice.

Reply:

This is a good question.

First of all, we have tried many times to synthesize the in-plane Na^+ -ion/vacancy disordering P2- $\text{Na}_{2/3}\text{Ni}_{1/3}\text{Mn}_{2/3}\text{O}_2$ with the intralayer TM honeycomb ordering. However, it is experimentally unreachable at current stage. Moreover, the DFT calculations disclose that the formation energy of LZZ type of Na^+ -ion/vacancy ordering is much lower than the other type ordering in P2- $\text{Na}_{2/3}\text{Ni}_{1/3}\text{Mn}_{2/3}\text{O}_2$ (Figure R9), which means the LZZ ordering is probably the thermodynamic

favorable configuration. Hence, we experimentally propose to obtain the $P2\text{-Na}_{2/3}\text{Ni}_{0.3}\text{Mn}_{0.7}\text{O}_2$ using tiny (slight composition) structural perturbation to minimize interference from other factors as shown in the Figures 1-2 of the manuscript. Although the preparation of $P2\text{-Na}_{2/3}\text{Ni}_{0.3}\text{Mn}_{0.7}\text{O}_2$ still has limited and controllable variables for comparisons to support the main content of this work, we have supplied more additional composition deviations from the optimal $\text{Na}_{2/3}\text{Ni}_{1/3}\text{Mn}_{2/3}\text{O}_2$ to support the observation as discussed in the Figures 1 and 2.

Figure R9 (Figure 4d in the revised manuscript) Energy differences of the in-plane Na^+ ion ordering in $P2\text{-Na}_{2/3}\text{Ni}_{1/3}\text{Mn}_{2/3}\text{O}_2$ structure, where the Mn^{4+} and Ni^{2+} ions described by a $(\sqrt{3} \times \sqrt{3})\text{-R}30^\circ$ supercell in transition metal layer.

Then, as for the DFT calculations, it is challenging to obtain an accurate atomic structure of $P2\text{-Na}_{2/3}\text{Ni}_{0.3}\text{Mn}_{0.7}\text{O}_2$ with TM (the stoichiometric Ni/Mn is 3/7) honeycomb ordering and Na slab disordering within present computation resources. Even constructing a disordered $P2\text{-Na}_{2/3}\text{Ni}_{1/3}\text{Mn}_{2/3}\text{O}_2$ structure is also a very difficult task for first principles calculations, where a significantly big supercell for DFT simulations always gets the converged results at a month-long time at current stage. Here, we have tried to enumerate the ground state structure from the relatively large supercell of the disordered $P2\text{-Na}_{2/3}\text{Ni}_{1/3}\text{Mn}_{2/3}\text{O}_2$ for DFT calculations under the current limited computing capabilities, that is, a $[4\sqrt{3}a \times 4\sqrt{3}b \times 1c]\text{-R}30^\circ$ -type superlattice in Wood's notation

which includes 352 atoms. Since it is difficult to consider all the paths of static diffusion in this large disordered supercell as mentioned in Comment 7, the AIMD simulation is performed at 800 K to disclose the Na^+ ion mobility difference induced by the disorder-to-order transition. Figure R10 shows the MSDs for Na-ions in P2- $\text{Na}_{2/3}\text{Ni}_{1/3}\text{Mn}_{2/3}\text{O}_2$ with Na^+ ion LZZ ordering and disordering at 800K. It can be seen that the MSDs of Na ion LZZ ordering is almost twice of it in disordering occupation, which means the enhanced Na-ion diffusivity after the structural disorder-to-order transition within the Na slabs.

Figure R10 (Figure 4e in the revised manuscript) Total MSDs for Na-ions in P2- $\text{Na}_{2/3}\text{Ni}_{1/3}\text{Mn}_{2/3}\text{O}_2$ with Na LZZ ordering and disordering at 800K.

However, due to the limited computational resources and the large disordered supercell, it is difficult to take all the available diffusion paths into account for the detailed CI-NEB analysis. Considering that the Na^+ ion disordering occupation can be decomposed into a random combination of all the available Na^+ ion orderings, that is, the honeycomb, chain and LZZ types here, it emerges the chance to clarify the disorder-to-order transition to accelerate the Na^+ ion diffusion in Na LZZ ordered structure. These discussions can be found in the following Comment 7 and Figure 4 (Page 20) of the revised manuscript or as follows:

“As a result, the MSDs of Na^+ ions by AIMD simulation in P2- $\text{Na}_{2/3}\text{Ni}_{1/3}\text{Mn}_{2/3}\text{O}_2$ with the all-possible

ordered configurations at 800 K are presented in Figure 4f. The noticeable migration of Na⁺ ions is captured in the structure with the LZZ ordering, while no directional movement of Na⁺ ions is found in the other two types of orderings. The probability density distribution (P(r)) of Na⁺ ions at 800 K provides more intuitive results, where the diffusion pathways form a 2D network with which all of the Na⁺ ion sites are connected via the adjacent vacancy as shown in Figure 4i, to support the BVEL results as shown in Figure 3b. In contrast, the Na⁺ ion honeycomb and chain orderings are probably immobilized in the energetically favorable crystal sites with the negligible ionic motions, as is similar case in P2-Na_{2/3}CoO₂ as shown in Figure S18.”

Next, in order to further verify the above results, the P3-Na_{2/3}Ni_{1/3}Mn_{2/3}O₂ polymorph can provide the possibility of directly comparing the Na⁺ ion diffusion kinetics (the structural difference is no preferential in-plane Na ordering in P3-Na_{2/3}Ni_{1/3}Mn_{2/3}O₂). In the P3 phase, all the Na 3a sites are face and edge-sharing with the adjacent TMO₆ octahedra as the equivalent crystal sites with equal formation energy in an asymmetric environment along the c axis, which is not easy to form Na⁺ ion LZZ ordering intrinsically. Hence, the precise synthesis of Na_{2/3}Ni_{1/3}Mn_{2/3}O₂ polymorphs opens the possibility of directly comparing the Na⁺ ion diffusion kinetics in P2/P3 phases with the exact stoichiometric ratios. Moreover, the widely used CI-NEB calculations only obtain the energy barrier of a specific static migration pathway. As an effective and powerful supplement, the AIMD simulations can depict the full physical picture of the ionic migration kinetics with the complex chemistry changes. Through the comparisons between P2/P3 Na_{2/3}Ni_{1/3}Mn_{2/3}O₂ polymorphs, the fast Na diffusion kinetics can be in-depth understood in P2- Na_{2/3}Ni_{1/3}Mn_{2/3}O₂ with Na slab LZZ ordering.

(5) I question the method in the paper that using only 20 candidate structures with the lowest Ewald energies can find the appropriate ground-state structures with the Na LZZ and TM honeycomb order. In contrast to the manuscripts, Meng et. al. (J. Chem. Phys. 2008, 128, 104708) used the complicated cluster expansion method to identify the Na LZZ order, and One et. al. (Phys. Rev. APPLIED 7, 064003 2017) enumerated 5000 structures to finally found the ground state structures. The authors should disclose more details of their methods for finding ground state structures.

Reply:

The details have been added in Page 7 of the revised manuscript as follows.

“To find the ground-state structure, approx. 300 Na⁺-ion/vacancy structural orderings with different Na_f ratios were enumerated²⁸ at the beginning. Then, the corresponding formation energies were calculated to set the lowest one as the ground state in P2-Na_{2/3}CoO₂, as is a similar case in P3-Na_{2/3}CoO₂. By fixing the arrangement of Na slab in the ground state structure, the Ni and Mn occupations were then generated with the honeycomb ordering in the transition metal layer that was derived from the NPD analysis. Afterwards, the above optimized crystal structures were transplanted to the P2- and P3-Na_{2/3}Ni_{1/3}Mn_{2/3}O₂ structures using the Ewald energy that is calculated in the Python Materials Genomics library²⁹ for further optimization. Finally, the one with the lowest total energy was set to the thermodynamically stable configurations of the corresponding P2 or P3 types. The [2√3a × 2√3b × 1c]R30°-type supercell containing 88 atoms for P2-type Na₁₆Ni₈Mn₁₆O₄₈ and the 3a × 3b × 1c supercell includes 99 atoms for P3-type Na₁₈Ni₉Mn₁₈O₅₄ were adopted for geometry optimizations and kinetics simulations. Moreover, the [4√3a × 4√3b × 1c]-R30°-type superlattice including 352 atoms was built to simulate the disordered Na⁺ ion occupations in P2-Na_{2/3}Ni_{1/3}Mn_{2/3}O₂. On the basis of the XRD/NPD results, the structure with [occ. (Na_f) = 1/12] and [occ. (Na_f) = 1/6] was set to further obtain the two disordered Na configurations using Pymatgen and Enumlib³⁰.”

(6) The P3 structure used in the computations is inconsistent with the experimental characterizations. Since the authors declare that Na is disorder in the P3-Na_{2/3}Ni_{1/3}Mn_{2/3}O₂, why did the authors used the P3 structure with order honeycomb Na (Figure S14) for computations? The authors are suggested to clarify the choice carefully, because the wrong use of the ground state structure may disqualify the whole computations.

Reply:

Thank you for your comments!

With the simulation method stated in the above Comment 5, the formation energies of different Na ion occupations in P3- $\text{Na}_{2/3}\text{Ni}_{1/3}\text{Mn}_{2/3}\text{O}_2$ structure can be calculated as shown in Figure R11 (Figure S21 in SI). It can be seen that there are two combined types of Na^+ ion orderings, and the energy differences of different Na arrangements are small enough (≤ 20 meV/f. u.) as shown in Figure R11 a below, which indicates that from a thermodynamic point of view, the co-contribution from honeycomb and chain type Na occupations probably represents a none preferential in-plane Na ordering in P3- $\text{Na}_{2/3}\text{Ni}_{1/3}\text{Mn}_{2/3}\text{O}_2$ structure, in good agreement with the experimental results. However, the ground state structure is further simplified within finite size supercell for the subsequently practical simulations. Since the overall disordering are due to the coexistence of multiple arrangements of similar energies ordering, the AIMD simulation is conducted in P3- $\text{Na}_{2/3}\text{Ni}_{1/3}\text{Mn}_{2/3}\text{O}_2$ structures with both honeycomb and chain ordering in this work to examine the diffusion mechanism.

As for the stable thermodynamic configuration, from a formation energy point of view, the van Hove correlation function and probability density in P3- $\text{Na}_{2/3}\text{Ni}_{1/3}\text{Mn}_{2/3}\text{O}_2$ (chain ordering) from AIMD simulations is presented in Figures R12 and R13 (Figure 5 in the revised manuscript), and the van Hove correlation function and probability density in P3- $\text{Na}_{2/3}\text{Ni}_{1/3}\text{Mn}_{2/3}\text{O}_2$ (honeycomb ordering) from AIMD simulations is provided in Figure R14 (Figure S25 in SI). The P3- $\text{Na}_{2/3}\text{Ni}_{1/3}\text{Mn}_{2/3}\text{O}_2$ (honeycomb ordering) exhibits much higher activation energy of 0.31 eV, and no connected channels are available within the Na slabs and the it seems difficult enough for one Na^+ ion diffusion to adjacent sites as shown in Figure S25 b. Of particular importance is that much lower activation energy is observed than in P2- $\text{Na}_{2/3}\text{Ni}_{1/3}\text{Mn}_{2/3}\text{O}_2$ with respect to that in both chain and honeycomb ordering P3- $\text{Na}_{2/3}\text{Ni}_{1/3}\text{Mn}_{2/3}\text{O}_2$. Hence, the above AIMD results fully support the conclusion.

These discussions have been added in Page 23 of the revised manuscript as follows.

“Next, the underlying reasons for the high Na^+ ion diffusivity are understood in P2- $\text{Na}_{2/3}\text{Ni}_{1/3}\text{Mn}_{2/3}\text{O}_2$ using AIMD simulations. Note that the AIMD simulations collect the statistic contributions of all diffusional events and provide an effective complement to the CI-NEB method, especially for the complex coordination environment of the targeted materials.³⁵ Based on the structural enumeration²⁹, the supercells of P3- $\text{Na}_{2/3}\text{Ni}_{1/3}\text{Mn}_{2/3}\text{O}_2$ are successfully built as shown in Figure S21, where there are

two assembled types of Na⁺ ion orderings in the supercell, and the energy differences of different Na arrangements are small enough ($\leq \sim 20$ meV/f. u.). This result indicates that from a thermodynamic point of view, the co-contribution from honeycomb and chain type Na occupations probably represents a none preferential in-plane Na ordering in P3- Na_{2/3}Ni_{1/3}Mn_{2/3}O₂ structure, in good agreement with the experimental results. However, the ground state structure is further simplified within finite size supercell for the subsequently practical simulations. The temperature dependent MSDs of Na⁺ ions in P2- and P3-Na_{2/3}Ni_{1/3}Mn_{2/3}O₂ (chain type) by AIMD simulations are shown in Figure S22 as a function of time.”

Figure R11 (Figure S21 in SI). (a) Formation energies of different Na ion occupations in P3- Na_{2/3}Ni_{1/3}Mn_{2/3}O₂ structure, where the P3-type supercell is constructed using a $3a \times 3b \times 1c$ -type lattice with 99 atoms (Na₁₈Ni₉Mn₁₈O₅₄). The 3H means three layers in honeycomb ordering, the 2H+C corresponds to two layers in honeycomb and one layer in chain orderings, the 1H+2C stands for one layer in honeycomb and two layers in chain orderings, and the 3C is all three layers in chain ordering. The simulation results indicate the chain type ordering is energy favourable structure. (b) Crystal environments of NaO₆ prism in P3 phases within the transition metal slabs. Atomic structure of P3-Na_{2/3}Ni_{1/3}Mn_{2/3}O₂ with the Na slab (c) honeycomb and (d) chain orderings.

Figure R12 (Figure S22 in SI). MSD of Na⁺ ions in (a) P2-Na_{2/3}Ni_{1/3}Mn_{2/3}O₂ (LZZ ordering) and (b) P3-Na_{2/3}Ni_{1/3}Mn_{2/3}O₂ (chain ordering) as a function of time at different temperatures.

Figure R13 (Figure 5 in the revised manuscript). Correlated sodium motions within Na slabs. (a) Arrhenius plot of Na^+ ion diffusivity in P2 and P3- $\text{Na}_{2/3}\text{Ni}_{1/3}\text{Mn}_{2/3}\text{O}_2$ (chain ordering) from AIMD simulations. The Na^+ ion diffusion pathway in (b) P3- $\text{Na}_{2/3}\text{Ni}_{1/3}\text{Mn}_{2/3}\text{O}_2$ within the ab plane from AIMD at 800 K, and the scalebar of the isosurface is set to 0.001. The self-part of the van Hove correlation function (G_s) for sodium in (c) P2- and (d) P3- $\text{Na}_{2/3}\text{Ni}_{1/3}\text{Mn}_{2/3}\text{O}_2$. The distinct part of the van Hove correlation function (G_d) for sodium ions in (e) P2- and (f) P3- $\text{Na}_{2/3}\text{Ni}_{1/3}\text{Mn}_{2/3}\text{O}_2$. Both G_d and G_s are functions of the average Na–Na pair distance (r) and time step after thermal equilibration at 800 K.

Figure R14 (Figure S25 in SI). (a) Transition metal plane with the honeycomb ordering of Mn^{4+} and Ni^{2+} ions in $\text{P3-Na}_{2/3}\text{Ni}_{1/3}\text{Mn}_{2/3}\text{O}_2$. (b) The Na^+ -ion diffusion pathway in $\text{P3-Na}_{2/3}\text{Ni}_{1/3}\text{Mn}_{2/3}\text{O}_2$ (Na slab honeycomb ordering) within the ab plane from AIMD at 800 K, and the scalebar of the isosurface is set to 0.001. (c) MSD of Na^+ ions in $\text{P3-Na}_{2/3}\text{Ni}_{1/3}\text{Mn}_{2/3}\text{O}_2$ (honeycomb ordering) as a function of time at different temperatures. (d) Arrhenius plot of Na^+ ion diffusivity in $\text{P3-Na}_{2/3}\text{Ni}_{1/3}\text{Mn}_{2/3}\text{O}_2$ (honeycomb ordering) from AIMD simulations. (e) The self-part of the van Hove correlation function (G_s) for sodium in $\text{P3-Na}_{2/3}\text{Ni}_{1/3}\text{Mn}_{2/3}\text{O}_2$. (f) The distinct part of the van Hove correlation function (G_d) for sodium ions in $\text{P3-Na}_{2/3}\text{Ni}_{1/3}\text{Mn}_{2/3}\text{O}_2$. Both G_d and G_s are functions of the average $\text{Na}-\text{Na}$ pair distance (r) and time step after thermal equilibration at 800 K.

(7) The direct and convincing evidence of the concerted migration to facilitate the diffusion is to perform the NEB calculations in both single and concerted ways, and compare their migration energies, as demonstrated in multiple studies (e.g., Nature Communications 8, 15893, 2017 and Solid State 34 / 45

Ionics, 371, 115767, 2021). The currently usage of the Van Hoff function to prove the concerted migration is far from enough to support the concerted migration mechanism, and it is deemed as a side evidence. Thus, the authors ought to perform the NEB calculations to prove the concerted migration can lower down the migration energies as illustrated in their table of content. The NEB migration energies can also be compared with those in P2-Na_{2/3}Ni_{0.3}Mn_{0.7}O₂, disorder P2-Na_{2/3}Ni_{1/3}Mn_{2/3}O₂ (suggested in comment 4) and P3-Na_{2/3}Ni_{1/3}Mn_{2/3}O₂ to finally prove the facilitation of Na ordering on diffusion.

Reply:

On the basis of the responses to major Comments 4-6, with the proposed Na ion disordering model (the disordering occupations mean the random combination of the all-possible types of Na⁺ ion orderings), the systematic CI-NEB and AIMD simulation were performed to examine the Na⁺ ion concerted migration mechanism in P2-Na_{2/3}Ni_{1/3}Mn_{2/3}O₂ with LZZ ordering.

Discussion has been added in Pages 19-22 of the revised manuscript as follows.

“As a matter of fact, the present consensus is apt to support that the Na⁺ ion diffusion in the O-type layered cathodes is harder than it in P-type ones due to the different coordination environment of Na⁺ ions in O type (octahedral) and P-type (prismatic, the larger open space for Na⁺ ion fast diffusion) materials.^{54, 55} The results here further point out the significant influence of the Na⁺ ion occupancy induced by a slight compositional variation on the ionic migration within the P-type layered cathodes.

Then, the AIMD and CINEB simulations are carefully conducted to explore the Na⁺ ion diffusion in both P2 and P3 phases. Taking the different lattice oxygen stacking sequences into consideration, the site stability of Na⁺ ion is calculated for the two prismatic sites, that is, the Na_f (2b) site face sharing with the TMO₆ octahedra, and the Na_e (2d) site edge sharing with the adjacent TMO₆ octahedra in P2-type layered oxides as shown in Figure 4a, b. Note that the practical simulation can be simplified to calculate the total energies of Na_e and Na_f sites in a simple P2-Na_xCoO₂ model to comparably understand the corresponding stability. The calculated total energies revealed that the Na_e site locates at a lower energy site of -168 meV/f. u. with respect to the Na_f site in P2-NaCoO₂ as shown in Figure 4c. Further structural enumeration in P2-Na_{2/3}CoO₂ indicate three types, viz., the honeycomb, chain

and LZZ type Na ion orderings in P2 phase. Figure 4d indicates that the formation energy of the intralayer Na⁺ ion LZZ ordering is lower than the Na⁺ ion honeycomb and chain orderings for approx. 152 and 100 meV/f.u. in P2-Na_{2/3}Ni_{1/3}Mn_{2/3}O₂, respectively. Then, the DOSs of P2-Na_{2/3}Ni_{1/3}Mn_{2/3}O₂ with three different in-plane Na⁺-ion/vacancy orderings are calculated as shown in Figure S17. The P2-Na_{2/3}Ni_{1/3}Mn_{2/3}O₂ with LZZ ordering exhibits an energy difference of 1.53 eV between the valence and conduction bands which is smaller than the other two orderings of 0.2 eV, which indicates the differences by the spatial distribution of Na⁺ ions. From a structural point of view, the structural stability of the P2 phase is not varying monotonously with the increased amount of Na⁺ ions at Na_e sites due to the Coulombic repulsion. Some Na⁺ ions will alternatively occupy the Na_f sites to balance and stabilize the complex interactions among the intralayer Na⁺ ions in the Na slabs, and the strong correlations between the NaO₆ and TMO₆ polyhedra.⁵⁶ In contrast, in P3-Na_{2/3}Ni_{1/3}Mn_{2/3}O₂ phase, all the Na 3a sites are face and edge-sharing with the adjacent TMO₆ octahedra as equivalent crystal sites with an equal formation energy in an asymmetric environment along c axis as shown in Figure 4a, b. The P2-Na_{2/3}Ni_{1/3}Mn_{2/3}O₂ with the disordered Na configurations (close to P2-Na_{2/3}Ni_{0.3}Mn_{0.7}O₂ as discussed above) were then enumerated computationally to examine the effect of Na⁺ ion ordering on facilitating its diffusion (supplementary Note 4). Figure 4e shows the total mean square displacements (MSDs) for Na-ions in P2- Na_{2/3}Ni_{1/3}Mn_{2/3}O₂ with Na LZZ ordering and disordering at 800 K. It can be seen that the MSDs of Na ion LZZ ordering is almost twice of it in disordering occupation, which means the enhanced Na-ion diffusivity after the structural disorder-to-order transition within the Na slabs. However, due to the limited computational resources and the large disordered supercell, it is difficult to take all the available diffusion paths into account for the detailed CI-NEB analysis. Considering that the Na⁺ ion disordering occupation can be decomposed into a random combination of all the available Na⁺ ion orderings, that is, the honeycomb, chain and LZZ types here, it emerges the chance to clarify the disorder-to-order transition to accelerate the Na⁺ ion diffusion in Na LZZ ordered structure. As a result, the MSDs of Na⁺ ions by AIMD simulation in P2-Na_{2/3}Ni_{1/3}Mn_{2/3}O₂ with the all-possible ordered configurations at 800 K are presented in Figure 4f. The noticeable migration of Na⁺ ions is captured in the structure with the LZZ ordering, while no directional movement of Na⁺ ions is found in the other two types of orderings. The probability density distribution (P(r)) of Na⁺ ions at 800 K provides more intuitive results, where the diffusion pathways form a 2D network with which

all of the Na^+ ion sites are connected via the adjacent vacancy as shown in Figure 4i, to support the BVEL results as shown in Figure 3b. In contrast, the Na^+ ion honeycomb and chain orderings are probably immobilized in the energetically favorable crystal sites with the negligible ionic motions, as is similar case in $\text{P2-Na}_{2/3}\text{CoO}_2$ as shown in Figure S18.

To further disclose the underlying reasons, the CI-NEB calculations are performed in the ordered $\text{P2-Na}_{2/3}\text{Ni}_{1/3}\text{Mn}_{2/3}\text{O}_2$. Five Na ion diffusion trajectories are considered under monovacancy migration, including three paths of single ion migration ($\text{Na}_e\text{-Na}_e$ and $\text{Na}_f\text{-Na}_f$) and two paths of multiple Na^+ ion concerted migration (Paths 3 and 5), whose diffusion trajectories are illustrated in Figure S19. The chain ordering structure is set as a reference due to its close formation energy with the LZZ structure and the higher occupancy of Na_f sites. Figure 4j shows the activation energies (E_{act}) of Paths 1, 2 and 4 are obviously lower than those of the Na^+ ions concerted migration (respective ~ 0.29 and 0.44 eV for two Na^+ ion along Paths 3 and 5). For the single ion migration, the Na^+ ion along Path 2 ($\text{Na}_f\text{-Na}_f$) encounters a lowest E_{act} of 0.09 eV, to imply the single migration dominated Na^+ ion diffusion within the chain ordering structure. Of special interest is that the single ion migration in LZZ ordering is not feasible since the Na^+ ions within the slab are highly correlated with each other. The movement of Na ion will inevitably lead to the involvement of Na ions nearby, which is actually the concerted migration of Na^+ ions as illustrated in Figures S20 and 4i-1. The E_{act} of $\text{Na}_e\text{-Na}_e$ migration with the assistance of one Na_f (Path 1) is estimated to be 0.05 eV, obviously lower than the Path 2 in chain ordering phase as shown in Figure 4j. The E_{act} of three Na^+ ion diffusion (Path 2) is also lower with regard to that of the single Na^+ ion migration. (Figure 4k) Therefore, the fast-ionic diffusion kinetics in LZZ ordering phase can be ascribed to the multiple ion concerted migration that is originated from its special ionic occupations. A conclusion remark is that when the Na slab LZZ ordering is disrupted, but the TM honeycomb ordering is kept in $\text{Na}_{2/3}\text{Ni}_{0.3}\text{Mn}_{0.7}\text{O}_2$ material, the interaction between Na ions is weakened significantly (degenerates to a random combination of other types of Na^+ orderings) and the effectively concerted Na^+ ion diffusion can not be formed by the intrinsic structural constraints.”

Figure 4. Na⁺ ion kinetics in P2/P3 Na_{2/3}Ni_{1/3}Mn_{2/3}O₂ layered oxides. (a, b) Crystal environments of NaO₆ prism in both P2 and P3 phases, where the local coordination of the Na_f (2b, faces-sharing) and Na_e (2d, edge-sharing) sites in P2-phase and the Na (3a, face and edge-sharing) site in P3-phase are shown. (c) Calculated formation energies of Na_e and Na_f sites in a simple P2-NaCoO₂ model. (d) Energy differences of the in-plane Na⁺ ion ordering in P2-Na_{2/3}Ni_{1/3}Mn_{2/3}O₂ structure, where the Mn⁴⁺ and Ni²⁺ ions described by a ($\sqrt{3} \times \sqrt{3}$)-R30° supercell in transition metal layer. (e) MSDs for Na-ions in P2- Na_{2/3}Ni_{1/3}Mn_{2/3}O₂ with Na⁺ ion LZZ ordering and disordering at 800 K, including the Disorder 1, [occ. (Na_f) = 1/6] and Disorder 2, [occ. (Na_f) = 1/12]. (f) Total MSDs for Na-ions in P2- Na_{2/3}Ni_{1/3}Mn_{2/3}O₂ with different occupations of Na at 800K. Isosurface of the probability density distribution P(r) of Na⁺ in P2- Na_{2/3}Ni_{1/3}Mn_{2/3}O₂ with (g) honeycomb, [occ. (Na_f) = 0] (h) chain, [occ. (Na_f) = 1/2] and (i) LZZ [occ. (Na_f) = 1/6] Na ion orderings within the Na slabs at 800 K, and the isosurface level is set to 0.001, where the green ball represents the Na_f site, and the yellow one represents the Na_e site. (j) Migration energy barriers for diffusion trajectories in layered P2-Na_{2/3}Ni_{1/3}Mn_{2/3}O₂ with chain

and (k) LZZ orderings. (i) Illustration of Path 1 in layered P2-Na_{2/3}Ni_{1/3}Mn_{2/3}O₂ with LZZ ordering in Figure 4k.

Minor comments:

(1) The equations used in the paper are often lacked for description of each factors, e.g., line 149 ρ and line 226.

Reply:

The factor descriptions of the equations have been supplemented in Pages 9 and 14 as follows.

In Page 9 “Here, the $\delta(\cdot)$ represents the one-dimensional Dirac delta function. The angular bracket is the ensemble average over the initial time t_0 . The $\mathbf{r}_i(t)$ stands for the position of the i^{th} ions at the time t . The N_d and r are the diffusing sodium ions in the unit cell and the radial distance, respectively. The ρ is the average number density which serves as the “normalization factor” in G_d .”

In Page 14 “the i_p represents the peak currents, n is the number of electrons per reaction species, the C_0 is the concentration of Na^+ in the lattice, A is the area of electrode, D is apparent Na^+ diffusion coefficient, v stands for the scan rate. The C_0 , A and n are roughly equivalent in these two P2 structures”

(2) The description in page 3 "the influence of the Na^+ -ion/vacancy ordering on Na kinetics for P2-type layered oxides has been less visited because of the significant challenges of finding a suitable material system to configure its contribution out solely. " may not be fair. A simple search can reveal related studies, e.g., Chemistry of Materials 26 (18), 5208-5214, 2014.

Reply:

The description in Page 3 is modified as follows.

“The influence of the Na⁺-ion/vacancy disorder-to-order transition on the Na kinetics seems to be an important content because of the significant challenges of finding a suitable material system to figure its contribution out in P2-type layered oxides.”

(3) The authors should justify the usage of the Hubbard U potential in their calculations. Their U parameters' choice is even different from their reference (Ref. 25). Authors are encouraged to calibrate their U parameters by aligning them to their experimental bandgap.

Reply:

Firstly, the values of Hubbard parameter U for Ni and Mn ions in this work are consistent with Nature Chemistry 8, 692–697 (2016) (Ref. 26 of our revised manuscript). Moreover, in Materials Project, the developer has calibrated U values for many transition metals of interest using the approach outlined in Wang et al.'s work (Solid State Ionics 166.1-2 (2004): 167-173.), where there seems to be a little change with respect to the early results because of the materials investigation progress. At present, the U values have been calibrated in transition metal oxides system.

Secondly, as shown in Figure R15, the band gaps of the P2 and P3 Na_{2/3}Ni_{1/3}Mn_{2/3}O₂ layered oxides are equivalent roughly revealed by the DOSs and the ultra-visible light absorption spectra, which plausibly proves the suitable values of the computational parameters in this study.

Figure R15. DOSs of (a) P2-Na_{2/3}Ni_{1/3}Mn_{2/3}O₂ with the Na slab LZZ ordering and (b) P3-Na_{2/3}Ni_{1/3}Mn_{2/3}O₂ with the Na slab chain ordering. (c) Experimental ultra-visible light absorption spectra of P2-Na_{2/3}Ni_{1/3}Mn_{2/3}O₂ and P3-Na_{2/3}Ni_{1/3}Mn_{2/3}O₂. (d) Dependence of $(\alpha(h\nu))^2$ vs. photon energy ($h\nu$) to check the optic band gaps.

(4) The authors should clarify their statement in page 9, i.e., Of particular interest is that the IE_{2g}/IA_{1g} ratio indicates the spatial distribution and occupation of Na⁺ ions to some extent. It is unacceptable to just provide the conclusion without any explanation or references.

Reply:

The broadened shoulder peaks at $\sim 640\text{ cm}^{-1}$ in P2-Na_{2/3}Ni_{0.3}Mn_{0.7}O₂ samples, is also detected in P2-Na_{2/3}Al_{1/24}Ni_{7/24}Mn_{2/3}O₂ and P2-Na_{2/3}Co_{1/6}Ni_{1/6}Mn_{2/3}O₂ (Figures S12 and S13). Through literature reviews (*Adv. Sci. Technol.* 74, 60-65 (2010); *Mater. Lett.* 135, 131-134 (2014)), these Raman vibrations where the Na ions participate are often smeared out because of the ionic mobility in lattice

and the related disorder distribution. As a matter of fact, this observation is line with the refined XRD/NPD data.

To clarify this, discussions have been supplemented in Page 12 as follows:

“Referring to the polarized Raman results by Qu *et al.*,³⁹ the Raman peak at 460 cm⁻¹ is identified as the E_{2g} mode, and another one at 576 cm⁻¹ is the A_{1g} mode of Na_xCoO₂, which corresponds to the Raman peaks at 478 cm⁻¹(E_{2g}) and 588 cm⁻¹(A_{1g}) with the apparent shifts in the as-prepared P2-Na_{2/3}Ni_{1/3}Mn_{2/3}O₂ and P2-Na_{2/3}Ni_{0.3}Mn_{0.7}O₂ samples here. Of particular importance is that the broadened shoulder peak at ~ 640 cm⁻¹ in P2-Na_{2/3}Ni_{0.3}Mn_{0.7}O₂ samples is also detected in P2-Na_{2/3}Al_{1/24}Ni_{7/24}Mn_{2/3}O₂ and P2-Na_{2/3}Co_{1/6}Ni_{1/6}Mn_{2/3}O₂ (Figures S12-13), which might connect with the Na ions disordered distribution or the possible sublattice formation.^{40, 41} This broadened Raman spectra is an indirect response to the structural disordering/ordering formations from the refined XRD/NPD results. After all, the Na slab ordering is disrupted, while the TM honeycomb ordering is kept in Na_{2/3}Ni_{0.3}Mn_{0.7}O₂ material.”

(5) In page 13, the authors concluded that electronic structures remain unchanged in P2-Na_{2/3}Ni_{0.3}Mn_{0.7}O₂ and P2-Na_{2/3}Ni_{1/3}Mn_{2/3}O₂ from the bandgap characterization, which is not convincing. The authors are suggested to perform the DFT calculations of the band structures to support their statement.

Reply:

On the basis of the responses to major Comments 4-6, it is almost impossible to obtain an accurate atomic structure of P2-Na_{2/3}Ni_{0.3}Mn_{0.7}O₂ with TM (the stoichiometric Ni/Mn is 3/7) honeycomb ordering and Na slab disordering for band structure calculations for comparison. As alternatives, the electronic structures of different Na ion occupations can be considered in the P2-Na_{2/3}Ni_{1/3}Mn_{2/3}O₂ system with TM honeycomb ordering. Figure R16 (Figure S17 in SI) shows the DOSs of P2-Na_{2/3}Ni_{1/3}Mn_{2/3}O₂ with the honeycomb [occ (Na_f) = 0], chain [occ (Na_f) = 1/2], and large zigzag (LZZ), [occ (Na_f) = 1/6] type Na ion occupations. The LZZ ordering exhibits the lowest gap of 1.53 eV, which

is 0.2 eV lower than the other two types of Na ion occupations. Moreover, the experimental ultra-visible light absorption spectra of $P2\text{-Na}_{2/3}\text{Ni}_{1/3}\text{Mn}_{2/3}\text{O}_2$ and $P2\text{-Na}_{2/3}\text{Ni}_{0.30}\text{Mn}_{0.70}\text{O}_2$ show that the optic band gap of both is roughly equivalent. Hence, the slight change in the calculated band gap should be strongly correlated with the occupations of the Na slab in $P2\text{-Na}_{2/3}\text{Ni}_{1/3}\text{Mn}_{2/3}\text{O}_2$.

In Page 15 of the revised manuscript, modifications are added as follows.

“Moreover, the optic bandgaps of $P2\text{-Na}_{2/3}\text{Ni}_{1/3}\text{Mn}_{2/3}\text{O}_2$ and $P2\text{-Na}_{2/3}\text{Ni}_{0.30}\text{Mn}_{0.70}\text{O}_2$ are equivalent roughly as shown in Figure S10, which means the probable exclusion of the electronic transport influence on the rate capability.”

Figure R16 (Figure S17 in SI). Atomic structure of $P2\text{-Na}_{2/3}\text{Ni}_{1/3}\text{Mn}_{2/3}\text{O}_2$ with three different in-plane Na^+ -ion/vacancy orderings. (a) Honeycomb [$\text{occ}(\text{Na}_f) = 0$], (b) chain [$\text{occ}(\text{Na}_f) = 1/2$], (c) large zigzag (LZZ) [$\text{occ}(\text{Na}_f) = 1/6$] types. The green ball represents Na_f and the yellow ball stands for Na_e . The DOSs of $P2\text{-Na}_{2/3}\text{Ni}_{1/3}\text{Mn}_{2/3}\text{O}_2$ with the (d) honeycomb, (e) chain and (f) LZZ type orderings.

(6) The content from line 295 to 300 is too puzzled to understand, please rephrase the sentences.

Reply:

The original content from line 295 to 300:

“Therefore, the Na LZZ ordering should be responsible for the good rate capability in P2 phase. This may be more prevailing in the materials design and optimization of layered cathodes for rechargeable batteries, in contrast to the consensus that the Na⁺ ion diffusion in the O-type phase is even harder than that in P-type one due to the different coordination environment of Na⁺ ions in O type (octahedral structure) and P-type (prismatic, the larger open space) materials.”

The rephrased text in Page 18 of the revised manuscript as follows.

“the present consensus is apt to support that the Na⁺ ion diffusion in the O-type layered cathodes is harder than it in P-type ones due to the different coordination environment of Na⁺ ions in O type (octahedral) and P-type (prismatic, the larger open space for Na⁺ ion fast diffusion) materials.^{54,55} The results here further point out the significant influence of the Na⁺ ion occupancy induced by a slight compositional variation on the ionic migration within the P-type layered cathodes.”

(7) In Figure 4i, the authors are suggested to change the scale of the y axis from the Ln based on e to log based on 10 to increase the readability as previous computational papers.

Reply:

Thanks for your carefully reading.

The scale of the y axis in original Figure 4i has been changed from Ln(D) to log (D), and the Arrhenius plot of Na⁺ ion diffusivity in P2 and P3-Na_{2/3}Ni_{1/3}Mn_{2/3}O₂ (chain type) from AIMD simulations has also been modified as follows (Figure R17).

Figure R17 (Figure 5a in manuscript). Arrhenius plot of Na^+ ion diffusivity in P2 and P3- $\text{Na}_{2/3}\text{Ni}_{1/3}\text{Mn}_{2/3}\text{O}_2$ (chain type) from AIMD simulations.

(8) In Figure 4i, the authors are suggested to add the error bar of each calculated diffusivity following the established method.

Reply:

As shown in minor Comment 7, the error bar of each calculated diffusivity has been added.

Thank you for all your comments.

Reviewer comments, second round

Reviewer #1 (Remarks to the Author):

The authors have addressed most of my previous concerns. However, it seems to me that the authors still failed to fully resolve the crystal structure. It should not be that much difficult to index the pattern and solve the structure, since it is already known that Ni-Mn are honeycomb ordered while Na-vacancy is zig-zag ordered. Though the current version can be published, I would recommend the authors to solve the structure. I will leave the decision to the editor.

Reviewer #2 (Remarks to the Author):

Based on the dedicated revision, the authors have provided satisfactory response to clarify my concern. Thus, I recommend the publication of the nice paper.

Reviewer #3 (Remarks to the Author):

This reviewer believes that the characterizations provided in the revised manuscript are sufficient to support the claims the authors have made, but they could improve the manuscript by providing indexing for the clear superstructure peaks based on the provided crystal structures in Table S2-S4.

Response to Reviewer #3 (Remarks to the Author):

This reviewer believes that the characterizations provided in the revised manuscript are sufficient to support the claims the authors have made, but they could improve the manuscript by providing indexing for the clear superstructure peaks based on the provided crystal structures in Table S2-S4.

Response:

Thanks for your comment.

The superstructure peaks have been indexed in Figure S3 and updated in the revised SI, based on the provided crystal structures in Tables S1-S4.

Details can refer to the following revised Figure S3:

Figure S3. Simulated NPD data of P2-type layered oxides with the transition metal (TM) honeycomb ordering, the Na slab large zigzag(LZZ) ordering and the coexistence of TM honeycomb and Na LZZ ordering using GSAS II. The superstructure peaks (TM honeycomb ordering and Na slab LZZ ordering) were clearly indexed based on the provided crystal structures in table S1 with the space group of $P3$.